# DeepETPicker: Fast and accurate 3D particle picking for cryo-electron tomography using weakly supervised deep learning

Guole Liu [1,2,3,8], Tongxin Niu[4,8], Mengxuan Qiu [1,2,3], Yun Zhu [5], Fei Sun [4,5,6,7] ✉ & Ge Yang [1,2,3] ✉

To solve three-dimensional structures of biological macromolecules in situ, large numbers of particles often need to be picked from cryo-electron tomograms. However, adoption of automated particle-picking methods remains limited because of their technical limitations. To overcome the limitations, we develop DeepETPicker, a deep learning model for fast and accurate picking of particles from cryo-electron tomograms. Training of DeepETPicker requires only weak supervision with low numbers of simplified labels, reducing the burden of manual annotation. The simplified labels combined with the customized and lightweight model architecture of DeepETPicker and accelerated pooling enable substantial performance improvement. When tested on simulated and real tomograms, DeepETPicker outperforms the competing state-of-the-art methods by achieving the highest overall accuracy and speed, which translate into higher authenticity and coordinates accuracy of picked particles and higher resolutions of final reconstruction maps. DeepETPicker is provided in open source with a user-friendly interface to support cryo-electron tomography in situ.

Structural biologists have traditionally followed a reductionist approach to handle cellular complexity, in which the molecular components of cells are isolated, purified, and then studied individually. Although this approach has been tremendously successful, it is also crucial to study the structures and functions of biological macromolecules in their native cellular environments[1]. Cryo-electron tomography (cryo-ET) provides a powerful tool for visualizing macromolecular complexes under native conformations at subnanometre resolutions and for revealing their spatial and organizational relationships[2]. This provides new mechanistic insights into key cellular processes and new possibilities for applications such as drug discovery. As biological samples are very sensitive to radiation damage, the native resolution of cryo-ET is limited to ~2–5 nm given the dose of imaging electrons that can be tolerated[3]. This resolution is insufficient for studying the structures and functions of macromolecular complexes. Subtomogram averaging (STA) is commonly used to obtain higher-resolution structures by aligning and averaging large numbers of particles of the same macromolecular complexes[4]. However, the manual picking of large numbers of particles is laborious and time-consuming. Automated tools for picking 3D particles from cryo-electron tomograms with high accuracy and efficiency are critically needed for high-resolution in situ structural biology.

[1]State Key Laboratory of Multimodal Artificial Intelligence Systems, Institute of Automation, Chinese Academy of Sciences, Beijing 100190, China. [2]National Laboratory of Pattern Recognition, Institute of Automation, Chinese Academy of Sciences, Beijing 100190, China. [3]School of Artificial Intelligence, University of Chinese Academy of Sciences, Beijing 100049, China. [4]Center for Biological Imaging, Institute of Biophysics, Chinese Academy of Sciences, Beijing 100101, China. [5]National Key Laboratory of Biomacromolecules, CAS Center for Excellence in Biomacromolecules, Institute of Biophysics, Chinese Academy of Sciences, Beijing 100101, China. [6]School of Life Sciences, University of Chinese Academy of Sciences, Beijing 100049, China. [7]Bioland Laboratory (Guangzhou Regenerative Medicine and Health Guangdong Laboratory), Guangzhou, Guangdong 510005, China. [8]These authors contributed equally: Guole Liu, Tongxin Niu. ✉e-mail: feisun@ibp.ac.cn; yangge@ucas.edu.cn

In addition to the intrinsically crowded cellular environment, at least two additional technical challenges are encountered when performing 3D particle localization and identification in cryo-electron tomograms. First, the total electron dose used in cryo-ET experiments is limited to minimize radiation damage, resulting in very low signal-to-noise ratios (SNRs) for the reconstructed tomograms[5]. Second, the tilt ranges of cryo-ET experiments are often restricted to ±60 degrees due to electron penetration depth limitations, which result in missing wedges in the reconstructed tomograms, causing structural distortions of macromolecular complexes in different orientations[6]. Overall, picking 3D particles from noisy and distorted tomograms of crowded cellular contents is substantially more challenging than picking 2D particles from cryo-electron micrographs for single-particle analyses.

To pick 3D particles for cryo-ET, both conventional and deep neural network (DNN)-based methods have been developed[7,8]. Among the conventional methods, template matching (TM)[9] and difference of Gaussians (DoG)[10] are widely adopted. In TM, the position and orientation of a predefined template that best matches the tomogram to be processed are determined by maximizing their cross-correlation. However, TM has several limitations, including its strong dependence on the predefined template, its requirement of manual threshold tuning for cross-correlation, and its high false-positive rates under low SNRs. DoG picks particles using a bandpass filter that removes high- and low-frequency components. However, it picks particles regardless of their classes, and its performance depends heavily on the tuning of its Gaussian filters for different datasets.

In recent years, DNN-based methods have become the state-of-the-art 3D particle picking approaches for cryo-ET[7,8,11–13]. For example, Faster-RCNN has been used to automatically locate and identify different structures of interest in tomograms in a slice-by-slice manner, but the 3D information between adjacent slices was not fully utilized[14]. To promote the development of 3D particle picking algorithms, the SHREC Challenge developed datasets of simulated cryo-electron tomograms to benchmark different particle picking methods[7,8,15]. The results showed that deep learning-based methods achieved much faster processing speed and much better localization and classification performance than conventional methods such as TM. In the SHREC2019 Challenge, DeepFinder achieved the best overall localization performance[13]. It uses a 3D-UNet to generate a segmentation voxel map and determines the positions of particles using a mean-shift clustering algorithm. In the SHREC2020 and SHREC2021 challenges, MC-DS-Net achieved the best overall classification performance by using a denoising and segmentation architecture. However, its model contains many parameters, imposing high hardware performance requirements. Moreover, MC-DS-Net is trained by real full masks of macromolecular particles, which are usually unavailable in real-world cryo-ET studies. In contrast, DeepFinder uses spherical masks for approximation[13]. These masks provide good performance for medium- and large-sized macromolecules but worse performance than real masks for small particles. Considering that real cryo-electron tomograms contain more complex intracellular environments than the simulated data used in the SHREC Challenges, the performances of those methods tested in the SHREC Challenges must be further validated on real experimental cryo-ET data.

Overall, despite the various automated particle picking methods developed for cryo-ET, their adoption in practice remains limited. This is mainly due to the limitations in their picking accuracy, processing speed and, for learning-based methods, manual annotation cost. In this study, to address the limitations, we develop a new deep learning-based method named DeepETPicker, which accurately and rapidly picks 3D particles from cryo-electron tomograms with a low training cost. It utilizes a 3D-ResUNet segmentation model as its backbone to distinguish biological macromolecules from their backgrounds in tomograms. The model training process of DeepETPicker requires only weak supervision using simplified labels and fewer training labels to attain performance comparable to that of competing methods, which reduce the cost of manual annotation. Fast postprocessing is performed on the generated segmentation masks to obtain the centroids of individual particles. To enhance the localization performance of DeepETPicker on small macromolecular particles, coordinated convolution and multiscale image pyramid inputs are incorporated into the architecture of the 3D-ResUNet model. To address the usual lack of real full masks of macromolecular particles in practice, different types of simplified weak labels are tested as replacements. To eliminate the negative influence of poor segmentation accuracy in edge voxels, a spatial overlap-based strategy is developed. Finally, to maximize the speed of particle picking, a customized lightweight model and GPU-accelerated pooling-based postprocessing are utilized.

When tested on simulated datasets from the SHREC2020 and SHREC2021 challenges, DeepETPicker achieves the highest overall processing speed and the best performance in both localization and classification. The performance of DeepETPicker is further verified on four real experimental cryo-ET datasets (EMPIAR-10045, EMPAIR-10651, EMPIAR-10499 and EMPIAR-11125). The results show that it outperforms the competing state-of-the-art methods by achieving higher authenticity and coordinates accuracy in picked particles and better resolution in final reconstructions. DeepETPicker is provided as open-source software with an easy-to-use graphical user interface (GUI)[16]. It will serve as a fast and accurate tool to support automated 3D particle picking for high-resolution cryo-ET in situ.

## Results
### Overview of DeepETPicker
The overall workflow of using DeepETPicker to pick 3D particles from tomograms (Fig. 1) consists of a training stage (Fig. 1a–d) and an inference stage (Fig. 1e–i). A tomogram is usually too large to be directly loaded into the DNN segmentation model for training because of memory constraints. Instead, it is partitioned into cubic volumes, which are often called subtomograms (Fig. 1a, c, e). During the training stage, given an input subtomogram, the parameters of the DNN segmentation model of DeepETPicker are adjusted to minimize the difference between its output and the ground truth, as defined by voxel-level annotation labels for the input subtomogram. Typically, more than 90% of the voxels are background voxels in experimental tomograms, and the proportion of macromolecular particles in volume is very small. To better segment particles of interest and to avoid over-segmenting the background, subtomograms centred on individual particles are extracted in the training stage. This strategy ensures that all annotated particles are used, and that each volume contains at least one particle. During the inference stage, every tomogram is scanned with a specific stride $s$ and a subtomogram size of $N \times N \times N$ (Fig. 1e). The trained DeepETPicker is used to process unseen subtomograms to produce voxel-level masks for individual particles. A GPU-accelerated pooling-based postprocessing operation is then performed to directly and rapidly identify particle centers (Fig. 1h). In this study, training and inference of DeepETPicker is performed on a single Nvidia GeForce GTX 2080Ti GPU.

DeepETPicker is provided as open-source software in Python with a friendly GUI (Supplementary Fig. 1) that integrates multiple functions, including preprocessing input tomograms, manually annotating particles, visualizing labeled particles, generating weak labels, and configuring parameters for particle picking. The visualization results can be adjusted via filtering and histogram equalization operations. Users can conveniently label particle centers or delete false labels. Different classes of particles in the same tomogram can be labeled simultaneously. The coordinates of labeled particles can be exported to files with different formats that are compatible with commonly used subtomogram averaging software.

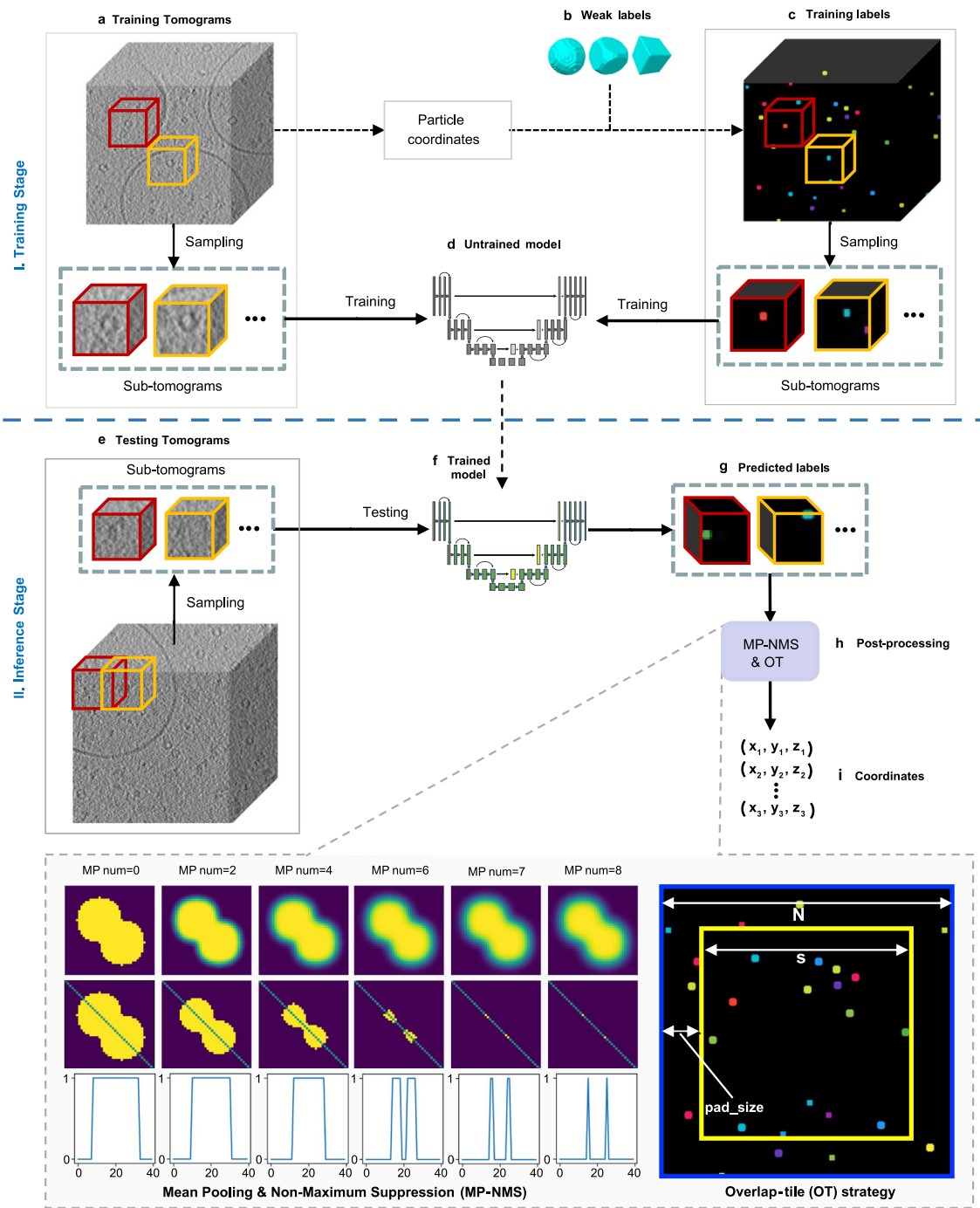

**Fig. 1 | Overall workflow of using DeepETPicker to pick particles from cryo-ET tomograms.** It consists of a training stage (**a-d**) and an inference stage (**e-i**). **a** Training tomogram: a reconstructed tomogram is partitioned into individual cubic volumes, referred to as subtomograms. **b** Weak labels: different types of simplified particle masks are generated to centre on manually annotated particle coordinates. **c** Training labels: the weak labels are assigned to their corresponding subtomograms. **d** Untrained model: a 3D-ResUNet model composed of a convolutional neural network with untrained parameters. **e** Testing tomogram: subtomograms partitioned from a new tomogram are used to test whether the trained model can accurately pick particles from unseen data. **f** Trained model. **g** Predicted labels: the trained model is used to predict voxel-level labels of the testing tomogram. **h** Postprocessing: mean pooling and nonmaximum suppression (MP-NMS) and overlap-tile (OT) operations are performed on the predicted labels. Specifically, an example of performing the MP-NMS operation on a 2D image with a size of 40 × 40 pixels is shown. **i** The positions of the picked particles are extracted after postprocessing.

## DeepETPicker achieves the best overall performance in picking particles from simulated tomograms

Under the very low SNRs of tomograms, it is difficult to generate full segmentation masks for macromolecular particles via manual annotation. To simplify the manual annotation process, three types of simplified masks (Ball-M, TBall-M, and Cubic-M) centred on manually labeled particle centers are generated (Supplementary Fig. 2a). For each type of simplified masks, their diameters can be set in different ways (Fig. 2b, c and Supplementary Table 2). Specifically, for the SHREC2021 dataset of simulated tomograms, the diameters of the simplified masks can be set to be proportional to the sizes of their corresponding real masks or as a constant value of 7 or 9. Utilizing simplified masks with constant diameters as training labels avoids the problem of class imbalance and simplifies the selection of the loss functions (Supplementary Methods).

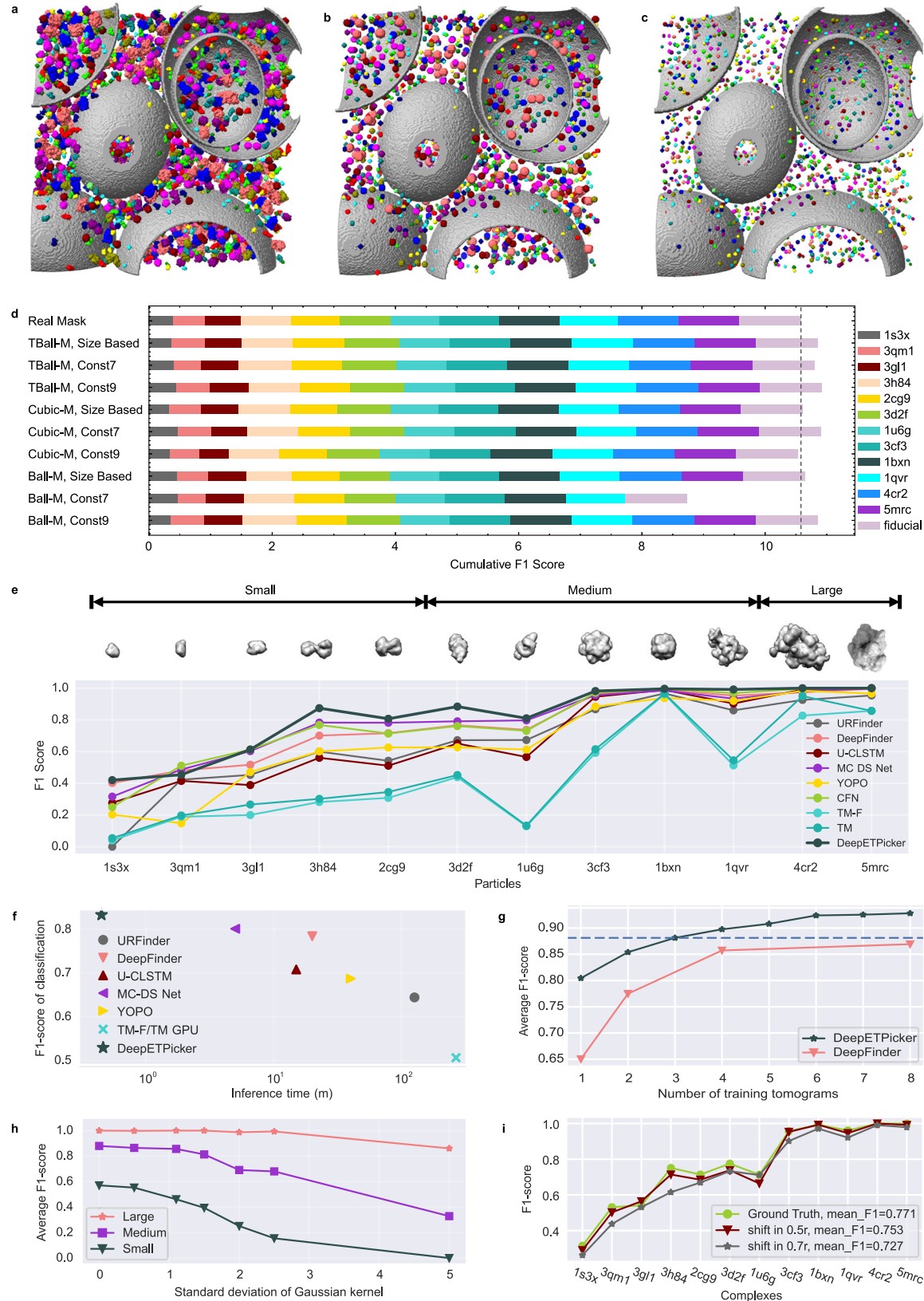

Compared to Cubic-M and Ball-M masks, TBall-M masks provide more stable and consistent localization and classification performance (Fig. 2d, Supplementary Table 3, and Supplementary Methods A.10). In addition, the 3D-RestUNet model trained by TBall-M achieves a mean F1-score that is 2% higher in absolute magnitude than that trained by real masks. This is likely because TBall-M may serve as noisy labels to replace real full masks, and the introduced label noise improves the generalization capability of the trained model on unseen datasets. Interestingly, TBall-M masks whose diameters are set in different ways have nearly the same localization and classification performance (Supplementary Table 3). Because simplified masks with a constant diameter are more convenient to set up in practice, the results in the

**Fig. 2 | Performance of DeepETPicker in comparison with that of competing methods on the SHREC2020 and SHREC2021 simulated datasets. a** Real full masks of macromolecular particles. Different colors are used to denote different classes of molecules. **b** Simplified TBall-M masks with diameters proportional to the sizes of their corresponding full masks. **c** Simplified TBall-M masks with a constant diameter $d = 7$. **d** Classification F1-scores achieved by DeepETPicker using real and different simplified/weak labels on the SHREC2021 dataset. Size-based: the diameter of each generated mask is proportional to the size of its corresponding real mask; Const 7 and Const 9: the diameters of the generated masks are fixed at 7 and 9, respectively. The dashed line shows the cumulative F1-score achieved by DeepETPicker when trained on real full masks. **e** Classification performance (measured in F1-scores) achieved on particles of different molecular weights (small: <200 kDa, medium: 200-600 kDa, large: >600 kDa): DeepETPicker versus other particle

picking methods reported in the SHREC2021 challenge. **f** DeepETPicker runs substantially faster and achieves substantially higher classification F1-scores than the competing particle-picking methods on the SHREC2021 dataset. **g** Classification performance (measured in F1-scores) under different numbers of training tomograms: DeepETPicker versus DeepFinder on the SHREC2020 dataset. The dashed line shows the average F1-scores of DeepETPicker when trained by three tomograms. **h** The influence of the SNR level on the classification performance of DeepETPicker for particles with different molecular weights from the SHREC2021 dataset. The noise levels under different Gaussian kernel $\sigma$ are 0.127 - 0.587 for $\sigma = 0$, 0.101 - 0.463 for $\sigma = 1.1$, 0.056 - 0.254 for $\sigma = 2$, and 0.026 - 0.110 for $\sigma = 5$. **i** The influences of different particle centre shifts (biases) on the classification performance of DeepETPicker. Source data are provided as a Source Data file.

remainder of this study are obtained using TBall-M masks with a constant diameter $d = 7$.

Precise particle centre localization is important for sub-tomogram averaging. Compared to other methods reported in the SHREC2021 challenge, such as URFinder, DeepFinder, U-CLSTM, MC-DS-Net, YOPO, TM-F, and TM, DeepETPicker achieves the best overall localization performance in terms of the TP, FP, FN, AD, precision, recall, and F1-score metrics (Supplementary Table 4). Specifically, compared to the best results obtained in the SHREC2021 challenge, DeepETPicker achieves a precision level of 0.958 (an increase of 8.9% in absolute magnitude), a recall value of 0.921 (an increase of 2.0%), an F1-score of 0.939 (an increase of 7.1%), and an AD of 1.15 (a decrease of 24.3%). For tomograms that contain a variety of macromolecular particles, accurate classification of these particles is critical, especially for different particles with similar molecular weights or similar geometries. DeepETPicker achieves the highest F1-scores on 10 types of macromolecular particles out of all 12 classes (Fig. 2e and Supplementary Table 5). The best mean F1-score among the competing methods is 0.801. DeepETPicker improves this mean F1-score by 3.75% in absolute magnitude. Overall, the classification F1-scores increase with increasing molecular weights, indicating that macromolecular particles with larger molecular weights are easier to pick, presumably because more voxels are occupied by larger particles in the same tomogram.

In the SHREC2021 challenge, the Multi-Cascade DS network (MC-DS-Net) achieved the best classification F1-score and the shortest inference time[15]. Compared to MC-DS-Net, DeepETPicker takes approximately 1/10 of its inference time and achieves better-picking performance (Fig. 2f and Supplementary Table 6). DeepETPicker achieves similar performance improvements over the methods in the SHREC2020 challenge[8] (Supplementary Fig. 5a, b). The customized lightweight and efficient architecture of 3D-ResUNet as well as the GPU-accelerated pooling-based postprocessing method, namely MP-NMP, are key factors that contribute to the performance of DeepETPicker.

The amount of annotated data used for training has a significant impacts on the picking performance of DNN-based models[13]. Compared to DeepFinder[13], DeepETPicker requires less training data to achieve the same level of performance on the SHREC2020 dataset (Fig. 2g). Specifically, the mean classification F1-score of DeepETPicker trained by 3 tomograms surpasses that of DeepFinder trained by 8 tomograms. When the classification F1-scores of particles with different sizes are plotted against the number of utilized training tomograms (Supplementary Fig. 5c), DeepETPicker shows a more pronounced classification performance advantage than DeepFinder for small particles.

We also examine the particle picking performance under different particle sizes combined with different tomogram noise levels. Specifically, we add different levels of Gaussian noise to the SHREC2021 dataset (Supplementary Table 7) and examine the influence of the noise level on the picking performance achieved under different

particle sizes. As the SNR decreases, the classification performance of DeepETPicker, measured by the F1-score, decreases (Fig. 2h). Moreover, the smaller the particle size is, the greater the decrease in the classification F1-score at lower SNR levels.

Manually labeling the particle centers in tomograms with extremely low SNRs inevitably introduces bias. For example, we calculate the Euclidean distance between the particle coordinates derived from manual picking and those obtained after refinement for EMPIAR-10499 (Supplementary Fig. 6). We find that 80% of the particles are less than $0.52r$ from the centre, where $r$ is the particle radius, and 90% of the particles are less than $0.625r$ from the centre. To better examine the impact of manual labeling bias on the particle picking results of DeepETPicker, we randomly add a shift between $0.5r$ and $0.7r$ to the particle centers of the SHREC2021 dataset. We find that the random shift has little impact on the picking performance of DeepETPicker for all complexes with different sizes (Fig. 2i). This indicates that DeepETPicker has good robustness against the localization bias induced by manual labeling.

We perform ablation studies on DeepETPicker and take 3D-UNet as the baseline to examine the contributions of the different customizations made to the 3D-RestUNet architecture in terms of improving picking performance (Supplementary Table 8). We find that adding residual connections (RCs) improves the mean F1-score of particle classification by 2%. Adding coordinate convolution (CC) and the image pyramid (IP) effectively improves the classification F1-scores obtained for small particles such as 1s3x and 3qm1 (Supplementary Methods). Data augmentation (DA) improves both the localization and classification performance of the model by substantial margins. The deduplication (DD) operation of removing the smaller particles among adjacent local maxima improves the localization F1-score by 1%. Finally, the overlap-tile (OT) strategy improves the F1-scores of both localization and classification by 5% and 4%, respectively, indicating its importance in the inference stage of DeepETPicker.

## DeepETPicker achieves the best overall performance in picking purified particles from real tomograms

We compare the performance of DeepETPicker with that of competing state-of-the-art methods in picking purified particles from two experimental datasets. The first dataset, EMPAIR-10045, consists of tomograms of purified *S. cerevisiae* 80 S ribosomes. It is widely used in the development of image processing algorithms for electron tomography[17]. The second dataset, EMPIAR-10651, consists of tomograms of purified T20S proteasomes from *Thermoplasma acidophilum*.

For EMPIAR-10045, we pick 80 S ribosome particles using DeepETPicker, crYOLO[18], DeepFinder[13] and TM[9] and examine the same and different particles picked by these methods in a pairwise fashion by calculating the intersection and difference sets of the picked particles (Fig. 3a, Supplementary Table 9 and Supplementary Movie 1). Based on the diameter of the 80 S ribosomes, we set $t_{dist} = 12$ to calculate the intersection and difference sets. We find that DeepETPicker picks true-

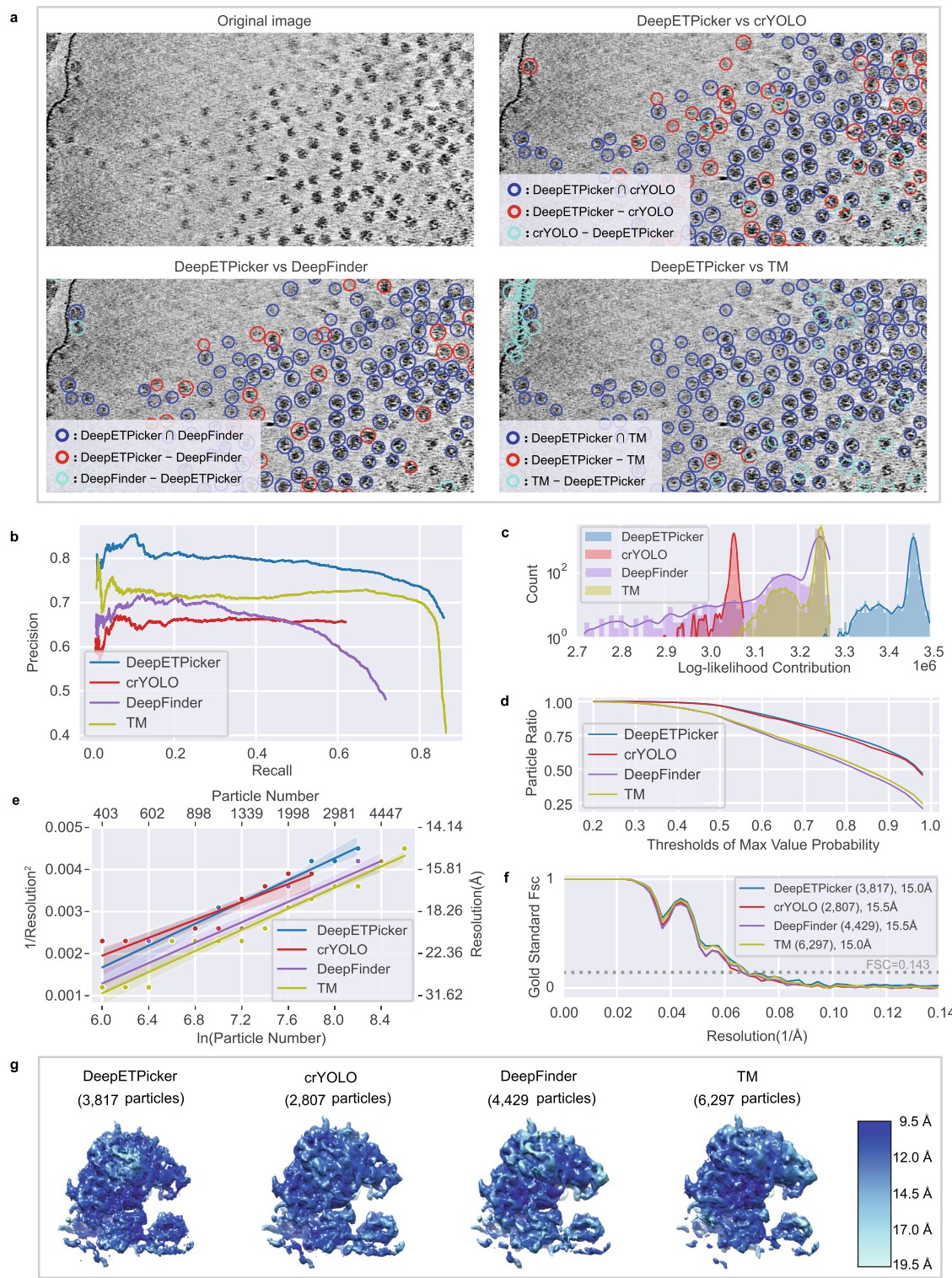

positive particles that are missed by the method reported in [17] as well as crYOLO and DeepFinder. Although TM picks many particles that are not selected by DeepETPicker, most of these particles are false positives (Fig. 3a).

Manual annotations are used to assess how closely an automated particle-picking method matches the manual particle picked by an expert. A comparison among the particles picked by the four selected methods (DeepETPicker, crYOLO, DeepFinder, and TM) and manual annotation is carried out via the precision and recall metrics (Supplementary Methods A.9). At a fixed recall, DeepETPicker achieves the highest precision, followed by TM, DeepFinder and crYOLO (Fig. 3b), indicating that the highest consistency with manual annotation is achieved by DeepETPicker. Furthermore, the maximal recall values of DeepETPicker and TM are substantially higher than those of Deep-Finder and crYOLO (Fig. 3b), indicating that more manually labeled particles are picked by DeepETPicker and TM. When the recall of TM reaches its maximum value, its precision decreases sharply, indicating that more false-positive particles are picked.

The authenticity and coordinates accuracy of the particles picked by these different methods are further examined based on the results

**Fig. 3 | Particle picking performance of DeepETPicker compared to that of the competing methods on the EMPIAR-10045 experimental dataset. a** Comparison between the particles picked by DeepETPicker and the three competing methods (crYOLO, DeepFinder, and TM). The original image is a result of performing Gaussian denoising and histogram equalization on the raw tomogram. Different colors are used to differentiate the same and different particles picked. The same particles picked by DeepETPicker and the competing method, i.e., those in the intersection sets of their picked particles, are shown in blue. The different particles picked by DeepETPicker and the other competing methods, i.e., those in the difference sets of their picked particles, are shown in red and cyan, respectively. **b** Precision-recall curves produced by different methods using manually picked particles as the reference. **c** Histogram of the log-likelihood contributions calculated by the RELION 3D auto-refinement method. Horizontal axis: log-likelihood contribution. Vertical axis: number of particles. **d** Particle ratio with a maximum value probability above a specific threshold calculated by the RELION 3D auto-refinement method. Horizontal axis: threshold of the maximum value probability. Vertical axis: ratio of the number of particles. **e** Rosenthal and Henderson B-Factor plot, which shows the relationship between the number of particles and the global resolution of the 3D reconstruction results. The translucent bands around the regression line denote the 95% confidence interval for the regression estimation. **f** FSC curves obtained by different particle picking methods after performing direct alignment and averaging. **g** Comparison of the local resolutions of the subtomogram averages obtained for budding yeast 80 S ribosomes using the particles picked by different methods (DeepETPicker, crYOLO, DeepFinder and TM). Source data are provided as a Source Data file.

of subsequent subtomogram averaging (Fig. 3c–g). For an objective comparison, no particle screening is performed during the subsequent alignment and classification processes because otherwise the measurements of the picked particles could be affected by the screening protocols used. We only set one class in the 3D classification step and perform 3D auto-refinement based on the shift and orientation parameters of the 3D classification method in RELION. Then, we plot the number of particles versus the corresponding calculated log-likelihood contribution (Fig. 3c). We find that the overall range of the log-likelihood contribution provided by the particles picked by DeepETPicker is consistently higher than those of the particles picked by crYOLO, DeepFinder, and TM. The same observation holds for the intersection and difference sets of the particles (Supplementary Fig. 7). Furthermore, we calculate the cumulative ratio of particles with the maximum probability higher than a threshold and plot the ratio versus the threshold (Fig. 3d). The cumulative ratio curves of DeepETPicker and crYOLO are close to each other but substantially better than those of DeepFinder and TM. Overall, the higher log-likelihood and the better cumulative statistics of the maximum value probability indicate that the particles picked by DeepETPicker are more authentic and accurate.

The assessment of the picked particles based on global resolution, local resolution, and B-factor measurements agrees with the assessment based on the log-likelihood distribution and the cumulative statistics of the maximum value probability (Fig. 3e–g). Specifically, the global resolutions of the reconstruction maps derived from the particles picked by DeepETPicker and TM are both 15.0 Å, which are slightly higher than those of the reported coordinates (15.1 Å), as well as those of DeepFinder and crYOLO (15.5 Å). Importantly, the map generated by particles picked by DeepETPicker exhibits the highest local resolution in comparison with those of crYOLO, DeepFinder, and TM (Fig. 3g). Based on the RH plots[19] (Fig. 3e), we observe that the set of particles picked by crYOLO gives the smallest slope, indicating that it has the highest B-factor. Although the slopes of the sets of particles picked by DeepETPicker, DeepFinder, and TM are similar, with the same number of particles, DeepETPicker achieves better global resolution than TM and DeepFinder.

The maps constructed from different particle datasets have similar global resolutions but different local resolutions and RH plots (Fig. 3e–g). We hypothesize that this is because of the differences in authenticity and coordinates accuracy among the different particles picked by different methods. To test this hypothesis, we perform subtomogram averaging on the particles in the difference sets and then compute their global resolutions. We find that particles picked by DeepETPicker but not by the other methods (crYOLO, DeepFinder, and TM) yield correct reconstruction maps (Supplementary Fig. 8) with global resolutions that are consistent with the RH resolution (Supplementary Fig. 9 and Supplementary Table 9), indicating that particles picked by DeepETPicker but missed by the other methods are true positives with authenticity similar to that of the true positives picked by these methods. The RH resolution is the theoretical resolution

estimated based on the RH plot. However, the particles picked by DeepFinder and TM but not DeepETPicker yield incorrect reconstruction maps (Supplementary Fig. 8) with global resolutions that are much worse than the RH resolution (Supplementary Fig. 9 and Supplementary Table 9), indicating that these particles are mostly false positives. Therefore, although the additional particles picked by DeepFinder and TM improve the SNRs of the half maps, i.e., reconstruction maps of the two independent halves of the datasets, and contribute positively to the FSC curve with an improved global resolution, they do not make a positive contribution to the RH plot and the local resolution.

To further examine the performance of different methods in picking particles with different shapes, we choose the T20S proteasome from EMPAIR-10651, which has a cylindrical shape. Following the same protocol as that of the analysis used above, we pick T20S proteasomes using DeepETPicker, crYOLO[18], DeepFinder[13] and TM[9] and calculate the same and different particles picked by these methods (Supplementary Fig. 10). According to the diameter of T20S proteasomes, we set $t_{dist} = 11$ for calculating the intersection and difference sets of the picked particles. Again, we find that DeepETPicker picks true-positive particles missed by crYOLO and DeepFinder (Supplementary Fig. 10a).

To further check whether this observation is true, a comparison between the particle picking results of different methods (DeepETPicker, crYOLO, DeepFinder and TM) and manual annotation is carried out via the precision and recall metrics (Supplementary Methods A.9). Overall, DeepETPicker and TM achieve comparable performance metrics, which are slightly better than those of DeepFinder and much better than those of crYOLO (Supplementary Fig. 10b). Furthermore, subtomogram averaging is performed to further check the authenticity and coordinates accuracy of the picked particles (Supplementary Fig. 10c, d). The global resolutions of the maps reconstructed from the particles picked by DeepETPicker, crYOLO, DeepFinder and TM are approximately 14.0 Å, 15.4 Å, 17.1 Å and 16.2 Å, respectively (Supplementary Fig. 10c). In agreement with the global resolution measurement, the map reconstructed from the particles picked by DeepETPicker shows more structural details and better local resolutions (Supplementary Fig. 10d).

## DeepETPicker achieves the best overall performance in picking particles in situ from real tomograms

Automated particle picking from real cryo-electron tomograms of cellular structures is critically needed for in situ structural biology. The crowded cellular environment poses a complex and challenging background for particle localization and identification, which is further compounded by the low SNRs of tomograms. Here, we first use the public cryo-ET dataset of native *M. pneumoniae* cells (EMPIAR-10499) to test the performance of DeepETPicker in picking ribosome particles in situ.

Following the same analysis protocol used above for the purified 80 S ribosomes of EMPIAR-10045, we pick 70 S ribosome

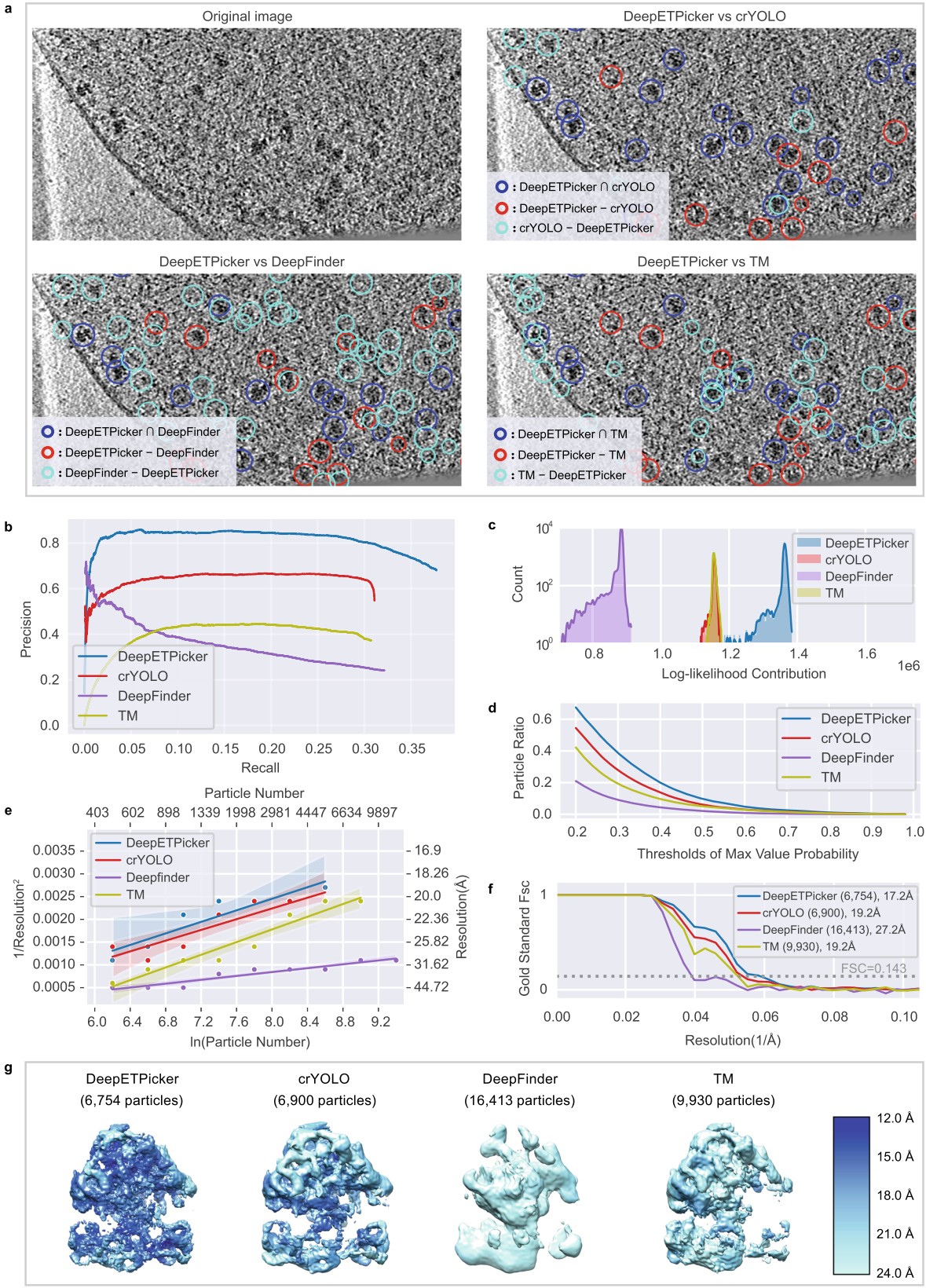

particles using DeepETPicker, crYOLO, DeepFinder[13], and TM[9] and calculate the same and different particles picked by these methods (Fig. 4a, Supplementary Table 10 and Supplementary Movie 2). According to the diameter of the 70 S ribosome, we set $t_{dist} = 12$ for calculating the intersection and difference sets of the particles. Again, we find that DeepETPicker can pick true-positive particles

missed by crYOLO and DeepFinder (Fig. 4a). Although DeepFinder and TM can pick particles not selected by DeepETPicker, these particles do not appear to be true positives upon initial visual inspection.

To further check whether this observation is true, a comparison between the particle picking results of different methods (DeepETPicker,

**Fig. 4 | Particle picking performance of DeepETPicker compared to that of the competing methods on the EMPIAR-10499 experimental dataset. a** Comparison between the particles picked by DeepETPicker and the other three competing methods (crYOLO, DeepFinder, and TM). The original image is a result of performing Gaussian denoising and histogram equalization on the raw tomogram. Different colors are used to differentiate the same and different particles picked. The same particles picked by DeepETPicker and the other competing methods, i.e., those in the intersection sets of their picked particles, are shown in blue. The different particles picked by DeepETPicker and the other competing methods, i.e., those in the difference sets of their picked particles, are shown in red and cyan, respectively. **b** Precision-recall curves produced by different methods using manual particles as the reference. **c** Histogram of the log-likelihood contributions calculated by the RELION 3D auto-refinement method. Horizontal axis: log-likelihood contribution. Vertical axis: number of particles. **d** Particle ratio with a maximum value probability above a specific threshold calculated by the RELION 3D auto-refinement method. Horizontal axis: threshold of the maximum value probability. Vertical axis: ratio of the number of particles. **e** Rosenthal and Henderson B-Factor plot, which shows the relationship between the number of particles and the global resolution of the 3D reconstruction results. The translucent bands around the regression line denote the 95% confidence interval for the regression estimation. **f** FSC curves obtained by different particle picking methods after performing direct alignment and averaging. **g** Comparison of the local resolutions of subtomogram averages obtained for *M. pneumoniae* 70 S ribosomes using particles picked by different methods (DeepETPicker, crYOLO, DeepFinder and TM). Source data are provided as a Source Data file.

crYOLO, DeepFinder and TM) and manual annotation is carried out using the precision and recall metrics (Supplementary Methods A.9). Overall, DeepETPicker achieves substantially higher precision than the other three methods under the same recall rate (Fig. 4b), indicating that the highest consistency with manual annotation is achieved by DeepETPicker. Subtomogram averaging is performed to further check the authenticity and coordinates accuracy of the picked particles (Fig. 4c–g). After calculating the log-likelihood contribution of each particle, the number of particles is plotted against the corresponding log-likelihood contribution (Fig. 4c). The range of the overall log-likelihood contributions of the particles picked by DeepETPicker again is substantially better than that of particles picked by crYOLO, DeepFinder, and TM. These conclusions also hold for their intersection and difference particle sets (Supplementary Fig. 11). Furthermore, the cumulative curves of the particle ratios are plotted against the maximum value probability (Fig. 4d). The cumulative ratio curve of DeepETPicker is consistently higher than that of crYOLO, TM, and DeepFinder. Therefore, the best log-likelihood contribution and cumulative statistics of the maximum value probability indicate that DeepETPicker picks particles in situ from tomograms more effectively and accurately than the other tested methods. This conclusion is further verified by the global resolution, local resolution, and B-factor measurements (Fig. 4e–g). The global resolutions of the maps reconstructed from the particles picked by DeepETPicker, crYOLO, DeepFinder, and TM are 17.2 Å, 19.2 Å, 27.2 Å, and 19.2 Å, respectively (Fig. 4f). In agreement with the global resolution measurement, the map reconstructed from the particles picked by DeepETPicker shows more structural details and better local resolutions (Fig. 4g). Although DeepFinder picks more particles, the final refinement step cannot converge into a correct map. Based on the RH plots (Fig. 4e), DeepETPicker achieves the highest global resolution using the same number of particles.

Following the same analysis protocol applied for the tomograms of purified 80 S ribosomes in EMPIAR-10045, we further analyse the same and different particles picked by DeepETPicker versus the other three methods by subtomogram averaging. We find that the particles picked only by DeepETPicker but not by the other methods (crYOLO, DeepFinder and TM) yield correct and plausible reconstruction maps (Supplementary Fig. 12) with global resolutions that are mostly consistent with the RH resolutions (Supplementary Fig. 13 and Supplementary Table 10). This indicates that the particles picked only by DeepETPicker are true positives with authenticity similar to that of common particles. However, all the different particles picked by the other methods and not by DeepETPicker yield incorrect reconstruction maps (Supplementary Fig. 12), with global resolutions that are substantially worse than the RH resolutions (Supplementary Fig. 13 and Supplementary Table 10). This indicates that the different particles picked by the other methods are mostly false positives. Furthermore, we inspect the particle distribution of the centre shifts for the same particles picked by DeepETPicker versus the other three methods. The shift range of the particles picked by DeepETPicker is smaller than that of other methods (Supplementary Fig. 14), indicating that the highest localization precision is achieved by DeepETPicker.

## DeepETPicker achieves the best overall performance in picking smaller particles in situ from real tomograms

The 80 S and 70 S ribosomes as well as the T20S proteasome studied above have molecular weights greater than 1 MDa. Particles with smaller molecular weights generally exhibit lower SNRs in tomograms, making particle picking more difficult. To test the performance of DeepETPicker in picking smaller particles in situ, we select a public cryo-ET dataset of *H. neapolitanus* alpha-carboxysomes (EMPIAR-11125)[20], whose molecular weight is 562 kDa.

Following the same analysis protocol used above for EMPIAR-10045, we pick alpha-carboxysome particles using DeepETPicker, crYOLO, DeepFinder[13] and TM[9] and calculate the same and different particles picked by these methods (Fig. 5). According to the diameter of *H. neapolitanus* alpha-carboxysomes, we set $t_{dist} = 7$ for calculating the intersection and difference sets of the picked particles. Again, we find that DeepETPicker can pick true-positive particles that are missed by crYOLO and TM (Fig. 5a). Although crYOLO also picks particles not selected by DeepETPicker, these particles do not appear to be true positives upon initial visual inspection.

To further check whether this observation is true, a comparison between the particle picking results of different methods (DeepETPicker, crYOLO, DeepFinder and TM) and manual annotation is carried out using the precision and recall metrics (Supplementary Methods A.9). At a fixed recall rate, DeepETPicker achieves the highest precision, followed by DeepFinder, TM and crYOLO (Fig. 5b), indicating that the highest consistency with manual annotation is achieved by DeepETPicker. DeepETPicker also achieves the highest recall, indicating that more manually labeled particles are successfully picked by DeepETPicker. Furthermore, we perform subtomogram averaging to further check the authenticity and coordinates accuracy of the picked particles (Fig. 5c–d). The global resolutions of the maps reconstructed from the particles picked by DeepETPicker, DeepFinder and TM are similar at ~7 Å (Fig. 5c). However, the particles picked by crYOLO fail to yield a correct reconstruction. In agreement with the global resolution measurement, the map reconstructed from the particles picked by DeepETPicker shows more structural details and better local resolutions (Fig. 5d). We also inspect the particle distribution of the centre shifts of the same particles picked by DeepETPicker versus the other three methods. The shift range of the DeepETPicker-picked particles is smaller than that of crYOLO and TM and is at the same level as that of DeepFinder (Supplementary Fig. 15), indicating that higher localization precision is achieved by DeepETPicker and DeepFinder.

## Discussion
Studying the high-resolution structures of macromolecular complexes in situ in their native cellular environments is at the forefront of contemporary structural biology. Cryo-electron tomography provides a powerful tool to achieve this goal. However, its application is limited by various technical bottlenecks, including the need to pick large numbers of macromolecular particles from tomograms at very low

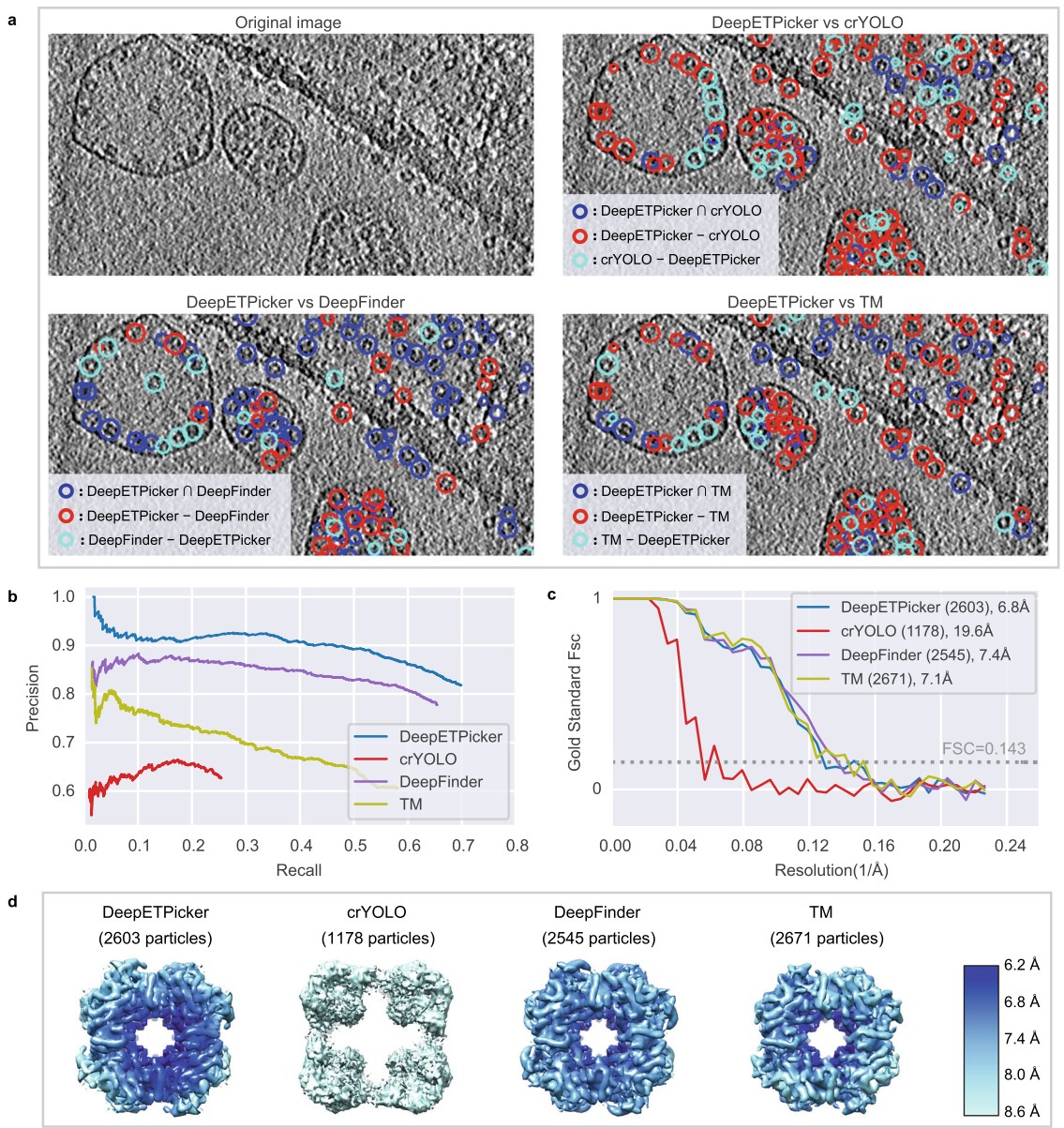

**Fig. 5 | Particle picking performance of DeepETPicker compared to that of the competing methods on the EMPIAR-11125 experimental dataset. a** Comparison between the particles picked by DeepETPicker and the other three competing methods (crYOLO, DeepFinder, and TM). The original image is a result of performing Gaussian denoising and histogram equalization on the raw tomogram. Different colors are used to differentiate the same and different particles picked. The same particles picked by DeepETPicker and the other competing methods, i.e., those in the intersection sets of their picked particles, are shown in blue. The different particles picked by DeepETPicker and the other competing method, i.e., those in the difference sets of their picked particles, are shown in red and cyan, respectively. **b** Precision-recall curves produced by different methods using manual particles as the reference. **c** FSC curves obtained by different particle picking methods after performing direct alignment and averaging. **d** Comparison of the local resolutions of the subtomogram averages obtained using particles picked by different methods (DeepETPicker, crYOLO, DeepFinder, and TM). Source data are provided as a Source Data file.

SNRs. In this study, we developed a new deep learning-based method, DeepETPicker, for automatic picking of 3D particles from tomograms with high accuracy and efficiency.

To address the lack of full segmentation masks for particles in real experimental tomograms, we designed three simplified masks, examined their performances, and found that the masks of TBall-M yielded the best and most stable results. We incorporated an overlap-tile strategy into the inference stage to avoid the negative influence caused by the poor segmentation accuracy achieved for edge voxels, which substantially improved the performance of both localization and classification (measured in F1-scores). We also proposed the MP-NMS operation for postprocessing to replace the clustering algorithms used

previously, which substantially improved the resulting inference speed. To help users pick particles from unlabeled tomograms and train DNN-based models, we developed a friendly graphical interface for DeepETPicker. Users can use this graphical interface to complete particle labeling, model training, and automatic particle picking with simple procedures.

We tested the performance of DeepETPicker and compared it with other state-of-the-art methods on two simulated datasets (SHREC2020 and SHREC2021) and we found DeepETPicker outperformed the competing methods with the highest average F1-score and the lowest computational time.

We also examined the performance of DeepETPicker on four experimental datasets (EMPIAR-10045, EMPIAR-10651, EMPIAR-10499, and EMPIAR-11125). We developed multiple particle metrics to compare the performance of DeepETPicker with that of other methods. We found that the particles picked by DeepETPicker consistently showed the best authenticity and coordinates accuracy with the highest log-likelihood contributions and the highest cumulative ratio of particles versus the maximum value probability, which was consistent with the observation that the particles picked by DeepETPicker produced reconstruction maps with the best global resolution, the best local resolution and the smallest B-factors. Although the assessment of reconstruction resolutions may be affected by the potential existence of specimen conformational heterogeneity, we found when comparing DeepETPicker with other methods such as crYOLO, DeepFinder[13] and TM, the particles not picked by DeepETPicker but selected by other methods generally failed to produce correct reconstructions. Therefore, the extensive analyses suggested that the accuracy and precision of the particles picked by DeepETPicker were substantially better than those of the other methods.

Overall, our study showed that DeepETPicker outperformed competing state-of-the-art methods on both simulated and real cryo-ET datasets. The results demonstrate the potential of DeepETPicker for applications in high-resolution cryo-ET studies in situ. In follow-up studies, we plan to incorporate particle orientation parameters into the framework of DeepETPicker, which will provide valuable information for the subsequent subtomogram averaging step. Furthermore, we plan to further optimize the classification performance of DeepETPicker on small particles.

## Methods

### Inclusion & ethics
We declare that our research complies with all relevant ethical regulations.

### Particle annotation using simplified labels
The supervised training of the DNN model of DeepETPicker requires pairs of subtomograms and their corresponding voxel-level masks/labels (Fig. 1a, c). Limited by the low SNRs and reconstruction distortion of tomograms, the manual voxel-level annotation of macromolecular particles is challenging and time-consuming. In this study, our goal is to identify particles rather than to obtain their full masks. To this end, we simplify the manual annotation process by only labeling the centers of particles, which is simple and efficient. Based on the annotations, three types of simplified masks centred on the labeled particles are generated as replacements for the real full masks, including Cubic masks (Cubic-M), Ball masks (Ball-M) and Truncated-Ball masks (TBall-M). Specifically, taking the centre of each particle as the origin, the corresponding simplified masks with sizes of $[2r+1, 2r+1, 2r+1]$ are generated as follows:

$$M = \{(x,y,z) \mid x,y,z \in [-r,r] \cap Z\} \tag{1}$$

$$mask_{cubic}(x,y,z)|_{(x,y,z) \in M} = c \tag{2}$$

$$mask_{ball}(x,y,z)|_{(x,y,z) \in M} = \begin{cases} c \ if \ \sqrt{x^2+y^2+z^2} < r \\ 0 \qquad\qquad otherwise \end{cases} \tag{3}$$

$$mask_{tball}(x,y,z)|_{(x,y,z) \in M} = \begin{cases} c \ if \ \sqrt{x^2+y^2+z^2} < \sqrt{2}\,r \\ 0 \qquad\qquad otherwise \end{cases} \tag{4}$$

where $c$ is the class index. TBall-M ensures that the generated TBall-M is sufficiently different from Ball-M and Cubic-M (See Supplementary Methods A.3 for further information). The diameter of each generated

mask is denoted as $d = 2r+1$, which should be no larger than the particle diameter. To ensure good particle picking performance, the diameter of the particles in the given tomogram should preferably be between 7-25 voxels. If the particle diameter is much larger than 25 voxels, proper binning operations can be used to keep the particle diameter within the recommended range. Examples of the three types of masks are shown in Fig. 1b and Supplementary Fig. 2a.

Compared to the real full masks of biological particles, these simplified masks can be seen as a class of weak supervision labels[21,22]. Subsequent experiments show that DNN segmentation models trained by these simplified labels can effectively segment/detect particles of interest in tomograms. See Supplementary Methods A.3 for a more detailed discussion on the rationale behind the selection of the simplified masks.

### Architecture of the 3D segmentation model
The DNN segmentation model of DeepETPicker, called 3D-ResUNet, adopts an encoder-decoder architecture (Supplementary Fig. 2b). Specifically, the residual connection idea from 2D-ResNet[23] is incorporated into 3D-UNet[24] to better extract features from tomograms. The 3D-ResUNet architecture has 3 downsampling layers in its encoder and 3 upsampling layers in its decoder. Three-dimensional transpose convolution is used in the decoder to upsample feature maps. An ELU[25] is used as the activation function to accelerate the convergence of the training process. To improve the localization of particles, coordinated convolution[26] and image pyramid inputs[27] are incorporated into 3D-ResUNet, which takes the voxel of each subtomogram as input and outputs $n$ probability scores for $(n-1)$ classes of structures of interest and the background, respectively, for each voxel. Coordinated convolution incorporates the spatial context of input images into the convolutional filters, while image pyramid inputs preserve features of input images at different resolution levels.

### Configuration for model training and validation
To improve the generalization capability of the segmentation model, data augmentation is used in the training stage. Specifically, the following transformations are performed on the training datasets: random cropping, mirror transformation, elastic deformation less than 5%, scaling in the range of [0.95, 1.05], and random rotation at angles within $[-15°, +15°]$. Training is performed using an AdamW optimizer[28] with an initial learning rate of $10^{-3}$ and a weight decay of 0.01. A multi-class extension of Dice loss[29] is used to calculate the difference between the predicted labels and ground-truth labels:

$$L_{Dice} = 1 - \frac{2 \sum_{i=0}^{N^3} p_i g_i + \varepsilon}{\sum_{i=0}^{N^3} p_i^2 + \sum_{i=0}^{N^3} g_i^2 + \varepsilon} \tag{5}$$

where $p_i \in \mathbb{R}^{N \times N \times N}$ denotes the labels predicted by the segmentation model, $g_i \in \mathbb{R}^{N \times N \times N}$ denotes the ground truth, and $\varepsilon = 10^{-8}$ is a small value added for numerical stability.

A generalized form of the F1-score, $F1_\alpha = \frac{2P \cdot R^\alpha}{P + R^\alpha}$, is used as a metric for model validation to place greater emphasis on model recall, where $\alpha$ is a hyperparameter. When $\alpha = 1$, $F1_\alpha$ becomes the F1-score. When $\alpha > 1$, the model with higher recall $R$ obtains a higher $F1_\alpha$. In this study, the hyperparameter $\alpha = 3$ is used.

### Postprocessing using mean-pooling non-maximum suppression (MP-NMS) and overlap-tile (OT)
The value of each voxel in the score map generated by 3D-ResUNet denotes its probability of belonging to a certain class, which is in the range of [0, 1]. A specific threshold $t_{seg}$ is selected to transform a score map into a binary map. A voxel whose value is below the threshold is labeled as 0 and otherwise as 1 so that a binary map is generated. Then,

the proposed MP-NMS operation, consisting of multiple iterations of mean pooling (MP) and one iteration of non-maximum suppression, is performed on the binary map as the initial input. An example of MP-NMS applied on a 2D binary image with a size of 40 × 40 pixels is shown in Fig. 1h. The first row shows the outputs of different iterations of MP operations performed on the binary image. After each MP operation, the voxels at mask edges are pulled closer to the voxel value of the background. As the number of MP iterations increases, all voxels of the mask are updated. Eventually, the binary mask is converted into a soft mask. The further a voxel in the mask is from the background, the larger its value. Each local maximum can be considered a candidate particle centre. The larger the local maximum, the higher the probability that it is a particle centre. MP-NMS can distinguish between the centers of multiple particles that partially overlap, as long as they have distinguishable features (Fig. 1h). Compared to clustering algorithms such as the mean-shift used in DeepFinder[13], the MP-NMS operation is substantially faster when accelerated using a GPU (Supplementary Table 6).

For an MP operation with a kernel size of $k \times k \times k$ and a stride of 1, the receptive field of each voxel after the $i^{th}$ iterations of MP operations is

$$RF_i = 1 + (k-1) * i \qquad (6)$$

To obtain the centroid of a particle with a diameter of $(2r+1)$, the receptive field $RF_i$ should be no smaller than the particle diameter. Thus, the minimum number of iterations of MP operations is $\lceil \frac{2r}{k-1} \rceil$, where $\lceil \cdot \rceil$ denotes the round-up operation.

To eliminate the negative influence of the poor segmentation accuracy achieved for edge voxels in subtomograms, an OT strategy is used in the inference stage. Taking the 2D segmentation case in Fig. 1h as an example and assuming that the image marked by the blue box is the output of the 3D-ResUNet model, only the centre region marked by the yellow box is considered during the inference stage to eliminate the poor segmentation of edge pixels. The size of the red box is determined using a hyperparameter termed 'pad_size'. Each tomogram is scanned with a specific stride $s$ and a subtomogram size of $N \times N \times N$ in the inference stage, where $s = N - 2 \cdot pad\_size$. Only the local maximum in the region of $[pad\_size : N - pad\_size, pad\_size : N - pad\_size, pad\_size : N - pad\_size]$ is retained.

To reduce background interference and avoid repetition during particle detection, two further postprocessing operations are performed. First, the local maxima below a threshold $t_{lm}$ are removed. Second, if the minimal Euclidean distance between two local maxima is lower than a specific threshold $t_{dist}$, the smaller local maximum is discarded.

## Metrics for picked particles

To compare the performance of DeepETPicker with that of other competing state-of-the-art methods, three performance metrics are used:

precision $P$, recall $R$, and the $F_1$-score $F1$[8,30], which are defined as follows:

$$P = \frac{TP}{TP + FP} \qquad (7)$$

$$R = \frac{TP}{TP + FN} \qquad (8)$$

$$F_1 = 2 \cdot \frac{P \cdot R}{P + R} \qquad (9)$$

where $TP$, $FP$ and $FN$ stand for true positives, false positives, and false negatives, respectively. For a particle with a radius of $r$, its predicted label is considered positive if the Euclidean distance from its

predicted centre to the ground truth is less than $r$. Otherwise, it is considered negative. To measure the localization accuracies of particle-picking algorithms, the average Euclidean distance (AD) from the predicted particle centre to the ground truth is calculated in voxels.

For real experimental datasets without ground truths, to compare the authenticity and coordinates accuracy of the particles picked by DeepETPicker and other competing state-of-the-art pickers, we used the B-factor, global resolution, local resolution, log-likelihood distribution, and maximum value probability for evaluation.

The B-factor of a set of particles is computed by the Rosenthal-Henderson plot (RH plot)[19], which shows the inverse of the resolution squared against the logarithm of the number of particles. A higher B-factor means that a lower inclination and a larger number of particles are needed to reach the same reconstruction resolution.

Another metric is the global 3D reconstruction resolution. By refining two models independently (one for each half of the data), the gold-standard Fourier shell correlation (FSC) curve is calculated[30–34] using the following formula:

$$FSC(k, \triangle k) = \frac{Real\left(\sum_{(k,\triangle k)} F_1(\boldsymbol{K}) F_2(\boldsymbol{K})\right)}{\left(\sum_{(k,\triangle k)} |F_1(\boldsymbol{K})|^2 |F_2(\boldsymbol{K})|^2\right)^{\frac{1}{2}}}, k = |\boldsymbol{K}| \qquad (10)$$

where $\boldsymbol{K}$ is the spatial frequency vector and $k$ is its magnitude. $F_1(\boldsymbol{K}), F_2(\boldsymbol{K})$ are the Fourier transforms of the reconstructions for the two independent halves of the datasets. The $FSC_{0.143}$ cut-off criteria[31] are used to calculate the global resolution.

In addition to the global resolution, the local resolution is another commonly used metric for evaluating the reconstruction map[35], which can be calculated in different ways by ResMap[35], MonoRes[36], DeepRes[37], etc. In this study, we use the ResMap algorithm implemented in RELION[17] to analyse the local resolution.

Furthermore, we propose two new metrics based on the Bayesian theory of subtomogram averaging implemented in RELION[38]. The approach of RELION aims to find the model that has the highest probability of being the correct one based on both the observed data and the available prior information. The optimization of *a posterior* distribution is called maximum *a posteriori* or regularized likelihood optimization. For a given dataset of picked particles, after *a posteriori* maximization, each particle is assigned two estimated parameters: one is called the log-likelihood to quantify its contribution weight to the final model, and the other is called the maximum value probability to quantify the accuracy of the particle parameter estimations (i.e., the orientation and the shift). The distribution statistics of the number of particles versus the log-likelihood and the cumulative statistics of the number of particles versus the maximum value probability are used in this study to evaluate and compare the particles picked by different pickers.

If the authenticity of the picked particles is worse, i.e., more false positive junk particles are picked, the SNR of the set of picked particles becomes worse. Then worse subtomogram averaging with reduced local and global resolutions is expected. Furthermore, a larger number of particles would be needed to reach the same reconstruction resolution. Thus a higher B-factor would be expected in case the local and global reconstruction resolutions may not be sensitive enough. It should be noted that if there is conformational heterogeneity in the specimen, the reconstruction resolutions, either local or global, may not be a good indicator to evaluate different pickers. More rigorous investigations using e.g., map inspection and 3D classification are needed.

More importantly, the authenticity and the coordinates accuracy of the picked particles can be assessed by the distributions of particle

log-likelihood and maximum value probability. For a picked particle that is false positive or has a large deviation from its true centre, a lower log-likelihood and a lower maximum value probability would be calculated to down-weight its contribution to the final subtomogram averaging. Therefore, a larger number of particles with higher log-likelihood and maximum value probability indicate better coordinates accuracy of the picked particles and better authenticity of the picked particle set.

## Comparison among the particles picked by different methods

To compare the sets of particles picked by two different methods, a duplication removal operation is performed to calculate their intersection and difference sets. Specifically, if the minimal Euclidean distance between two particles is lower than a specific threshold $t_{dist}$, which is normally set to half of the diameter of the particle, the two particles are considered the same. The intersection set contains particles picked by both methods, whereas the two difference sets contain particles picked by one but not the other method. For example, if we denote the set of particles picked by method $A$ simply as $A$ and the set of particles picked by method $B$ simply as $B$, the particles in the intersection set $A \cap B$ are picked by both method $A$ and method $B$. The particles in the difference set $A - B$ are picked by $A$ but not by $B$, whereas the particles in the difference set $B - A$ are picked by $B$ but not by $A$. Further explanations and illustrations of the intersection and difference sets are given in Supplementary Fig. 3.

## Datasets used for performance benchmarking

The performance of DeepETPicker is benchmarked on both simulated and real cryo-ET tomograms from six datasets: SHREC2020, SHREC2021, EMPIAR-10045, EMPIAR-10651, EMPIAR-10499, and EMPIAR-11125. The DeepETPicker hyperparameters used for these datasets are summarized in Supplementary Table 1. For each of the four experimental EMPIAR datasets, the overall workflow is to manually label the selected particles, use the labeled particles for model training, and, finally, use the trained model to pick particles from all testing tomograms. Detailed information on how each dataset is partitioned for training, validation, and testing is provided in the Supplementary Methods.

SHREC2020 is a dataset of simulated cryo-ET tomograms[8]. It consists of 10 tomograms of cell-scale volumes. Each tomogram contains 12 classes of protein particles that vary in size, structure, and function. Ranked by their molecular weights from small to large, the Protein Data Bank (PDB) codes of the 12 classes of protein particles are 1s3x, 3qm1, 3gl1, 3h84, 2cg9, 3d2f, 1u6g, 3cf3, 1bxn, 1qvr, 4cr2 and 4d8q. Tomograms 0 to 7 are used for training, tomogram 8 is used for validation and hyperparameter optimization, and tomogram 9 is used for testing. DeepETPicker takes tomogram voxels as its inputs. For each voxel, it outputs 13 probability scores that correspond to the 12 protein classes and the background, respectively.

SHREC2021 is another dataset of simulated cryo-ET tomograms[15]. Compared to SHREC2020, some major updates were made to the simulation process. Gold fiducial markers and vesicles were added to provide realistic additional challenges. SHREC2021 consists of 10 tomograms of cell scale volumes. Each tomogram contains 12 classes of protein particles that vary in size, structure, and function. Ranked by their molecular weights from small to large, the PDB codes of the 12 classes of protein particles are 1s3x, 3qm1, 3gl1, 3h84, 2cg9, 3d2f, 1u6g, 3cf3, 1bxn, 1qvr, 4cr2 and 5mrc. Tomograms 0 to 7 are used for training, tomogram 8 is used for validation and hyperparameter optimization, and tomogram 9 is used for testing. DeepETPicker takes tomogram voxels as its inputs. For each voxel, it outputs 15 probability scores that correspond to the 12 protein classes plus vesicles, gold fiducial markers, and the background, respectively.

EMPAIR-10045 is a real experimental cryo-ET dataset. It contains 7 tomograms of purified *S. cerevisiae* 80 S ribosomes[17]. Each tomogram

contains an average of 445 manually picked particles. The original tomogram and manually picked particle coordinates are contained in the subdirectory of the EMPIAR entry. Based on the aligned tilt series, ICON[39] is used to reconstruct tomograms with better contrast for particle picking (Supplementary Fig. 4a). To reduce the computational cost and to increase the SNR, the tilt series are downsampled 4× before performing ICON reconstruction so that the diameter of the 80 S ribosome in the final tomogram is ~23-24 voxels. For particle picking and performance comparisons, four different methods are chosen, including DeepETPicker, crYOLO[18], DeepFinder[13], and TM[9]. TM is performed by Dynamo[40] with a reference map from EMDB entry EMD-0732 low-pass filtered to 60 Å (see the tutorial http://wiki.dynamo. biozentrum.unibas.ch/w/index.php/Walkthrough_for_template_ matching). A total of 150 manually labeled particles are used for training and validation of DeepETPicker, crYOLO and DeepFinder (See Supplementary Methods A.8). The tutorials (http://cryolo. readthedocs.io/en/stable/tutorials/tutorial_overview.html and https:// deepfinder.readthedocs.io/en/latest/tutorial.html) provided for crYOLO and DeepFinder are followed for model training and particle picking. Based on the obtained coordinates of ribosome particles, subtomograms are directly extracted from the original tomograms. Subtomogram averaging is performed (Supplementary Fig. 4a) by following the reported protocol using the same parameters[17], including CTF estimation, particle extraction, 3D classification (with one class only) and 3D autorefinement. The CTF model of each particle is generated using RELION scripts.

EMPAIR-10651 is a real experimental cryo-ET dataset of cylindrical T20S proteasomes from *Thermoplasma acidophilum*[41]. It contains 3 tomograms of purified T20S proteasomes. Based on the aligned tilt series contained in the subdirectory of the EMPIAR entry, tomo3d is used to reconstruct the tomograms (Supplementary Fig. 4b). To reduce the computational cost and increase the SNR, the tilt series are downsampled 4× before performing tomo3d reconstruction so that the diameter of the T20S proteasome in the final tomogram is ~21 voxels. Similar to EMPAIR-10045, DeepETPicker, crYOLO[18], DeepFinder[13], and TM[9] are chosen for particle picking and performance comparisons. TM is performed by Dynamo[40] with a reference map from EMDB entry EMD-12531 low-pass filtered to 60 Å. A total of 142 manually labeled particles are used for training and validation of DeepETPicker, crYOLO and DeepFinder (See Supplementary Methods A.8). Similar to EMPAIR-10045, the model training and particle picking processes of crYOLO and DeepFinder are performed following the respective tutorials provided. Based on the obtained coordinates, subtomograms are extracted from the original tomograms. Then, subtomogram averaging is performed in RELION 2.1.0 (Supplementary Fig. 4b), including CTF estimation, particle extraction, 3D classification (with one class only) and 3D auto-refinement. The CTF model of each particle is generated using RELION scripts.

EMPIAR-10499 is a real experimental cryo-ET dataset of native *M. pneumoniae* cells treated with chloramphenicol[42]. In this study, we focus on picking 70 S ribosome particles from these in situ tomograms. Ten tomograms (TS_77, TS_78, TS_79, TS_80, TS_81, TS_82, TS_84, TS_85, TS_87 and TS_88) from this dataset are selected for particle picking and verification purposes (Supplementary Fig. 4c). CTF estimation and motion correction are performed on the original movie stacks using Warp 1.0.9[43], and the tilt series, as well as the tilt angle files, are imported into IMOD 4.9.12[44] for tilt alignment and tomogram reconstruction using the weighted back-projection algorithm with a radial filter cut-off of 0.35 and a fall-off of 0.05. To reduce the computational cost and increase the SNR, the reconstructions are downsampled 4× so that the diameter of the 70 S ribosome in the final tomogram is ~23-24 voxels. Again, DeepETPicker, crYOLO[18], DeepFinder[13], and TM[9] are chosen for particle picking and performance comparisons. TM is performed by Dynamo with a reference map from EMDB entry EMD-21562 low-pass filtered to 60 Å. A total of

117 manually labeled particles are used for training and validation of crYOLO and DeepETPicker, and 703 particles are used for training and validation of DeepFinder (See Supplementary Methods A.8). Finally, RELION 2.1.0 (Supplementary Fig. 4c) is used to perform subtomogram averaging, including CTF estimation, particle extraction, 3D classification (with one class only) and 3D auto-refinement. The CTF model of each particle is generated using RELION scripts. The local resolution is directly calculated using RELION 2.1.0.

EMPIAR-11125 is an experimental cryo-ET dataset of *H. neapolitanus* alpha-carboxysomes[20]. Three stacks (CB_02, CB_29, CB_59) are available from its EMPIAR entry for particle picking and verification purposes (Supplementary Fig. 4d). CTF estimation and motion correction are performed on the original movie stacks using Warp 1.0.9[43]. Tilt alignment is performed using Dynamo (https://github.com/alisterburt/autoalign_dynamo). To reduce the computational cost and increase the SNR, the reconstructions produced by Warp are downsampled 8× and then used for particle picking so that the diameter of the alpha-carboxysome in the final tomogram is -13 voxels. Again, DeepETPicker, crYOLO[18], DeepFinder[13], and TM[9] are chosen for particle picking and performance comparison purposes. TM is performed by Dynamo with a reference map from EMDB entry EMD-27654 low-pass filtered to 60 Å. A total of 571 manually labeled particles are used for training and validation of crYOLO, DeepETPicker and Deep-Finder (See Supplementary Methods A.8). Due to memory constraints, the final reconstructions are performed using 2× downsampled data in Warp. Then, RELION 3.1 beta is used for the subsequent subtomogram averaging step, including 3D classification (with one class only) and auto-refinement (Supplementary Fig. 4d).

### Statistics & reproducibility
In this study, 10 independent random replicates were produced on SHREC2021 datasets. Randomized two-sample t-test (rndttest2) as well as nonparametric ranksum test were performed for statistical comparison. For real tomograms, sample sizes were determined through experimental validation. No statistical method was used to pre-determine sample sizes. DeepETPicker was evaluated across 4 experimental datasets, see detailed information in Supplementary Table 19. No data were excluded from the analysis. The material for reproducing the results within Figures and Supplementary Figs. is available in the Source Data file and the tutorials at https://github.com/cbmi-group/DeepETPicker.

### Reporting summary
Further information on research design is available in the Nature Portfolio Reporting Summary linked to this article.

## Data availability
All relevant data supporting the key findings of this study are available within the article and its Supplementary Information files. The simulated tomogram dataset SHREC2020 is available from the website of the SHREC2020 challenge [https://www.shrec.net/cryo-et/2020/]. The simulated tomogram dataset SHREC2021 is available from the website of the SHREC2021 challenge [https://www2.projects.science.uu.nl/shrec/cryo-et/]. The experimental tomogram dataset of purified *S. cerevisiae* 80 S ribosomes is available from EMPIAR under accession number EMPIAR-10045. The experimental tomogram dataset of purified T20S proteasomes is available from EMPIAR under accession number EMPIAR-10651. The experimental tomogram dataset of *M. pneumoniae* cells is available from EMPIAR under accession number EMPIAR-10499. The experimental tomogram dataset of *H. neapolitanus* alpha-carboxysomes in situ is available from EMPIAR under accession number EMPIAR-11125. For template matching, reference maps are available from EMDB entry: EMD-0732, EMD-12531, EMD-21562, and EMD-27654. Source data are provided with this paper.

## Code availability
The code and user documentation for DeepETPicker are openly accessible at https://github.com/cbmi-group/DeepETPicker[16]. Detailed tutorials are provided on each step of particle picking for single-class and multi-class examples.

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

## Acknowledgements

This study was supported in part by research grants from the Strategic Priority Research Program of the Chinese Academy of Sciences (No. XDB37040102 to F.S., No. XDB 37040402 to G.Y.), National Natural Science Foundation of China (No. 31925026, No. 61932018 to F.S., No. 91954201, No. 31971289 to G.Y.), National Key Research and Development Program (No. 2021YFA1301500 to F.S.), Chinese Academy of Sciences (No. 292019000056 to G.Y.), and University of Chinese Academy of Sciences (No. 115200M001 to G.Y.). We thank Teng Wang for valuable discussion and technical assistance on docker deployment for DeepETPicker.

## Author contributions

F.S. and G.Y. designed the project and oversaw overall planning and execution. G.L. designed and implemented the DeepETPicker method and its graphical user interface. T.N. and M.Q. carried out the computational experiments. Y.Z. provided technical advice on method development and computational experiments. G.L., T.N., G.Y., and F.S. wrote the paper with inputs from all authors. G.Y. and F.S. secured research funding.

## Competing interests

The authors declare no competing interests.
