## [Peer Review File · Nature Communications]

DeepETPicker: Fast and accurate 3D particle picking for cryo-electron tomography using weakly supervised deep learningReviewer #1 (Remarks to the Author):

The authors introduce a simple, but very effective, algorithm to identify macromolecules in electron tomograms. This is still an open problem in which any new contribution is welcome and expected to have an important impact on daily practice. The algorithm is technically sound and well-described. The authors present a very extensive analysis of its performance on various datasets, both simulated and experimental.

This reviewer finds that the manuscript can be published unaltered.

Reviewer #2 (Remarks to the Author):

This manuscript introduced DeepETPicker, a novel deep-learning-based method for performing particle picking in 3D cryo-electron tomograms along with a user-friendly GUI. This method only requires the particle center locations to be provided for training purposes and does not require the voxel level segmentation maps of tomograms, unlike the other cryo-ET particle picking methods. The method is based on U-Net architecture like similar methods but incorporated several other modules including mask formation, MP-NMS, OT, etc. The authors validated their approach on two simulated benchmark datasets and two experimental ribosome tomogram datasets. DeepETPicker resulted in overall better performance in picking than the baseline and produced a higher resolution structure when combined with subtomogram averaging. By requiring only particle centers for training and by providing an interactive GUI for selecting particle centers, DeepETPicker stands out from other methods as a practically useful particle picking approach.

Despite showing much promise, there are several major and some minor concerns regarding the paper which are mentioned below:

Major concerns:

1. The authors mentioned tuning hyperparameters as a drawback of reference-free DoG picking. However, their method requires several hyperparameters (t_g , t_{seg} , t_{dist}) as well. For t_g , they have always used 0.368. But is there any rationale behind choosing this value? The paper did not mention clearly how t_{seg} was chosen. The paper would benefit from discussing how the choice of hyperparameter values affects the final picking results.
2. For the evaluation of simulated SHREC cryo-ET tomograms, the proposed method outperformed the baseline methods in several metrics, such as Precision, Recall, and F1 score. The improvement ranged between 2%-9%. However, in terms of another commonly used metric, AD (average distance), its performance was 24.3% less than that of the baseline method. Is there any justification behind this?
3. One of the lucrative selling points of the paper is that it only requires labeling the particle centre coordinates to prepare training data. The authors claimed that labeling the centers of the particles is easy and efficient, which was certainly the case for the tomograms they used for validation. But will it be the case for other tomograms as well, particularly when the particle size is small and the SNR of tomograms is lower? It would be nice to see a discussion on the ease of selecting particle centers with respect to the particle size and the noise level in the tomogram.

Minor concerns:

1. It is not clear how the method benefited from the use of coordinated convolution and image pyramid inputs. A detailed ablation study would be helpful.
2. The authors referred to supplementary figures 2c and 2d from the main manuscript. If supplementary figure 2 is very important to understand the manuscript, then I would recommend moving it to the main manuscript.

3. A table corresponding to Figure 2f would better help the reader to understand the relative improvement by DeepETPicker for simulated tomograms.

4. Can the method lead to any significant biological insight? Apparently, it does seem to directly present any new biological insight.

Reviewer #3 (Remarks to the Author):

This manuscript describes a supervised deep learning method for identifying macromolecule species in cryo-ET images. Evaluation is performed on a simulated benchmark dataset, as well as two experimental datasets. Comparison to two state-of-the-art methods (template matching and DeepFinder) is provided.

I recognise that the authors have put considerable efforts into the evaluation. Also, the method shows encouraging scores on the simulated benchmark. However, I consider that additional efforts are necessary for publication in Nature Communications. A major revision is needed. Below are the main reasons:

- I notice that evaluation on experimental focus on ribosomes, which among the macromolecule species is the easiest to identify. While these results validate the ability of DeepETPicker to identify ribosomes, I consider that the evaluation should also be focused on macromolecules that are representative of contemporary research in structural biology, i.e. macromolecules with a molecular weight < 1MDa.
- While this method presents novelties, I find that the method is an incremental work compared to DeepFinder. This is not something bad in itself, but it should be mentioned (please refer to [1] for more details). Some aspects of the method are unclear. Also, the way the authors implemented one of their method features, the "gaussian mask" is quite odd, and I have a number of doubts (see [2]).
- I have several concerns about the evaluation protocol on the experimental datasets (see [3]).

[1] Similarities and differences to DeepFinder

[1.1] What is similar to DeepFinder:

- use of a 3D semantic segmentation network to detect macromolecules
- same sampling strategy: sampling of 3D patches (subtomograms) centered on annotated coordinates, so that each patch contains at least 1 macromolecule. This is to mitigate the under-representation of the object classes w.r.t. to background class. This process is key for a successful training.
- use of "weak labels", i.e. placing spheres at annotated macromolecule positions
- overlap-tile strategy in inference stage
- use of Dice loss

[1.2] What is different:

- the network architecture.
- post-processing of the produced segmentation maps to obtain macromolecule coordinates. DeepFinder uses spatial meanshift clustering, and DeepETPicker uses mean-pooling and non-maximum suppression.

The way you applied NMS here is unclear to me (even with suppl Fig2. D). Could you please elaborate? Also, see [4.2] for a remark on how you implemented NMS.

[2] About the use of "weak labels"

[2.1] Implementation of "gaussian masks"

The authors compare 3 ways of generating weak labels: cubic, ball (i.e. sphere) and gaussian. The benefits of using a gaussian function is that it produces smooth values (decreasing from the center), which can be useful for generating score-maps to be used as training targets (which would need a regression loss function, e.g. MSE). But the authors binarize the gaussian function so that the training targets correspond to segmentation maps (and proceed to use Dice loss). But a binarised 3D gaussian (as described in Eq. (3) of the manuscript) is... a sphere. Even though in Suppl. Fig. 2 (a) the masks "Bal-M" and "Gau-M" appear to be different, actually "Gau-M" is still a sphere whose radius is larger than the used observation window. So the authors present experiments to demonstrate that their gaussian masks are better than spherical masks (the latter being already used in cryoET, see ref [13] of the manuscript), but it turns out both are identical. Could you please explain?

[2.2] Interrogation about the use of spherical masks

You advocate for the use spherical/"gaussian" masks, which is a good approximation for macromolecules whose shape is sphere-like. How do you plan to handle the case of elongated macromolecules (for ex. double-capped 26S proteasome, EMD-3932)? In this case the overlap between the real shape (cylindrical) and the modelled shape (spherical) is reduced, and hence the label noise is increased, which may prevent the training from converging.

[2.3] Remark on how the "weak label" strategy is presented

In the abstract, the authors write: "Training of DeepETPicker requires only weak supervision by simplified Gaussian-type labels, which reduce the burden of manual annotation of tomograms under very low signal-to-noise ratios"

Also, on p.5, l.12: "Model training of DeepETPicker requires only weak supervision using simplified labels, resulting in substantially reduced cost of manual annotation".

The authors make it seem as if their "simplified labels" will save biologists a lot of time in the annotation process, who would otherwise have to produce voxel-wise annotations. In reality, producing manually voxel-wise annotations for macromolecules is an impossible task, because of very weak contrast and blurry edges. So in fact, without the "weak annotation strategy", their method would simply be impracticable, which is quite different to the authors statement.

[3] About the evaluation protocol for the EMPIAR datasets

[3.1] Concern about the described subtomogram averaging pipeline

According to p.13 l.28, the authors use 3D classification in their subtomogram averaging procedure, which is very disturbing. This means that the authors apply post-classification to the particle picking method outputs. They should compare the raw output of the methods, else they do not measure the method performances, but the performance of a [method + post-classification] pipeline. This basically invalidates all subtomogram averaging results on EMPIAR-10045 and EMPIAR-10499.

[3.2] Questions about your evaluation criteria

The term "quality of particles" is mentioned around 15 times in the text, but this term is vague and the authors do not define what they mean by that. For all experiments involving subtomogram averaging, they put a considerable amount of effort measuring this "quality", see for example: p.10, l.23: "For real experimental datasets without ground truth, we used B-factor, global resolution, local resolution, and log-likelihood distribution to evaluate and compare the quality of particles picked by DeepETPicker and other competing state-of-the-art methods".

What exactly do you mean by particle "quality"? The SNR of the subtomogram? How does measuring this "quality" help quantifying the performance of a particle picker? Shouldn't a good particle picker detect all target macromolecules, regardless of the SNR amount (even the most noisiest ones)?

As a reminder, the resolution of a subtomogram average quantifies the structural homogeneity of picked particles. The resolution is low when the average is noisy (e.g. particle count is low), or when the average is blurry (e.g. particles are not similar enough). Structural heterogeneity occurs

when: (i) picked particles are not of the same macromolecule species (i.e. false positives), and (ii) picked particles are of the same macromolecule species, but with different structural conformations.

Here is an example to illustrate my point. The 26S proteasome has 2 assembly states: single-capped, double-capped. Including both states in the average will produce blur at the cap locations (i.e. resolution is low at these locations). As for the detector, it did not make any mistakes, as it picked proteasomes correctly (regardless of their assembly state). So in this case, the resolution of obtained subtomogram average does not reflect well the performance of the particle picker.

I think that it would be beneficial to the paper to discuss how the resolution of an obtained subtomogram average is correlated to the performance of a particle picker (and under which conditions).

[3.3] When comparing DeepETPicker to template matching and DeepFinder on the EMPIAR datasets, the authors do not explain how they chose the score thresholds of the last two methods. Yet the performance of TM and DeepFinder depends greatly on the chosen threshold, and therefore it should be mentioned when making a comparison.

[3.4] For all subtomogram averages you show, please indicate the number of particles used to compute it, as poor resolution may be due to low particle count. I noticed that you give some particle counts in Suppl. Table 9, however it would be easier to interpret the averages if you also display these values on the figures.

[3.5] According to Suppl. Table 9, we have:

- Subset $TM \cap \text{DeepETPicker}$: nb_of_particles=3662, resolution=19.2 Å

- Subset $TM - \text{DeepETPicker}$: nb_of_particles=6333, resolution=19.2 Å

Both subset averages have the same resolution of 19.2Å, which suggests that both subsets have a good true positive rate. This also suggests that DeepETPicker missed 6333 ribosomes, which is a lot. Could you please comment on this?

[3.6] For the intersection of two sets A and B, we should have $A \cap B = B \cap A$. Could you please explain why this is not the case in Suppl. Tables 8 and 9? For ex. in Suppl. Table 9, $\text{DeepETPicker} \cap TM$, and $TM \cap \text{DeepETPicker}$ contain 3618 and 3662 particles, respectively (i.e. a difference of 44 particles!).

[3.7] Why don't you compare the detections to the manual annotations (via precision, recall, F1-score)? It is true that manual annotations are not a ground truth, but it gives an indicator of how close the particle picking method is to what an expert has picked.

[3.8] How do you explain that DeepFinder performs well on EMPIAR-10045 and poorly on EMPIAR-10499? Given that DeepFinder has been shown to have state-of-the-art performance for detecting ribosomes (and even differentiate between two binding states of the ribosome) in cellular tomograms (*Chlamydomonas reinhardtii* cells)?

[3.9] p.13, l.22: "We selected 150 particles from manual annotation to train DeepETPicker, DeepFinder and TM."

Did you make sure that the train and test particles do not originate from the same tomograms? When sampling 3D patches centered around annotated particles, there are good chances that the patches also includes additional neighboring particles. If the split into train and test sets is not done carefully, there is a risk of overlap between train and test sets (i.e. a test patch may contain particles from train set), which results in metrics being overestimated.

[3.10] p.14, l.13: "We manually picked 117 particles to train crYOLO and DeepETPicker, and 703 particles to train DeepFinder"

Why are you using more particles for DeepFinder? Does this mean that the train and test sets are not the same for all compared methods? Please describe clearly the train, valid and test sets for the EMPIAR-100045 and EMPIAR-10499 datasets.

[4] Other remarks

[4.0] The authors announce their software to be open-source, but the github page they link to does not contain the source code. However it does contain a docker image, which I was not able to execute (this might be due to my configuration). I could nevertheless access their code through the docker image, but it would be better to be able to visualize the code in github directly. This would allow users to estimate the quality of the code more easily. For example, it seems like the authors repository does not follow the standard organisation of a python package (as needed for distribution on PyPI). Following these standards allow for better reproducibility and reusability of the method. I am well aware that this is not a condition for publishing, please consider this as a strong recommendation.

[4.1] p.8 l.11: "The value of each voxel in the segmentation maps [...] denotes its probability score of belonging to a certain class, and the score is in the range of [0,1]"
You seem to confuse the terms "segmentation map" and "score map". A segmentation map has integer values in the range of [0, n_classes]. A score map has float values in the range of [0,1].

[4.2] p.8, l.17: "Then the proposed MP-NMS operation [...] are performed on the binary image to obtain local maxima" and then p.8, l.24: "The larger the local maximum, the higher the probability that this is a particle"

I find it quite odd to estimate local maxima on binary images. Even more so if you use this value for non-maximum suppression, given that all local maxima will have the same value: 1.

[4.3] p.8, l.26: "Compared with clustering algorithm such as mean-shift used in DeepFinder, our proposed MP-NMS operation is substantially faster when accelerated using a GPU"

Here you should point to Suppl. Table 6. Which brings me to the next point: this table is supposed to compare computing times, however for each method a different GPU is used (sometimes even multiple GPUs). Therefore these numbers are not comparable, and you should not call this table a "comparison". Also, the computing times reported in Fig.2 (e) do not mention which hardware was used.

[4.4] p.16, l.16: "DeepETPicker achieves similar performance improvements over methods in the SHREC2020 challenge. This is because DeepETPicker utilizes customized lightweight and efficient architecture in its 3D-ResUNet segmentation model as well as a GPU accelerated pooling-based post-processing method"

The authors claim that their architecture is more lightweight than competing methods in the SHREC2020 challenge, and that this is one of the reasons why DeepETPicker performs better. However, when I compare their architecture (21 convolutional layers) on Suppl. Fig. 2 to the architecture (15 convolutional layers) used in DeepFinder (one of the competitors in SHREC2020), it seems to me that the latter has less parameters (taking into account the number and the size of the filters).

[4.5] p.9, l.23: "Specifically, the following transformations are performed on the training datasets: random cropping, mirror transformation, elastic deformation less than 5%, scaling in the range of [0.95, 1.05], and random rotation in angles withing [-15°, 15°]"

In my opinion, data augmentations should not change the data distribution in ways that are not desirable. Here is what I mean by that: firstly, the main clue for identifying a macromolecule species is its shape, which includes its chirality. Therefore, should you be using mirror operations? Secondly, cryo electron tomograms have an anisotropic resolution. If you use random rotations, the orientation of mentioned anisotropy is changed. Should your model be invariant to the anisotropy orientation?

[4.6] p.15, l.23: "Precise localization of particle centers is crucial for sub-tomogram averaging"
While it is certainly desirable to obtain a precise localization, it is not "crucial" for sub-tomogram

averaging. Having small localisation errors is fine, as in subtomogram averaging, not only the macromolecule orientation is estimated, but also a spatial shift.

[4.7] p.2 l.7: "adoption of automated particle-picking methods remains limited because of their limitations in [...] training cost"

This is in my opinion an odd wording. How can a training cost be "limited"? With respect to what?

[4.8] p.4, l.9: when speaking of the DoG picker, you should mention that this picker is not class-specific, unlike all other methods mentioned. DoG picks particles regardless of their class.

[4.9] p.10, l.14-18: "TP is shorthand for 'true positive', namely a positive particle is predicted to be positive ..."

The authors provide detailed definitions for the terms true positives, false positives and false negatives. In my opinion this is not necessary, and it will make you save some lines.

[4.10] p.19, l.3: "Therefore, although the additional particles by DeepFinder and TM improves the SNR of the whole dataset [...]"

This statement is bizarre. How does a particle picking method improve the SNR of the "whole dataset"? I assume that by "whole dataset" you mean the tomogram set. You are not performing denoising, so what do you mean by "improve the SNR"?

[4.11] p.19, l.12: "The crowded cellular environment adds additional background to make particle picking [...] more difficult".

This sentence is clumsy. There is no background that is being added (the "amount" of background stays the same). With a crowded cellular environment, the background becomes rather more complex/challenging.

A Point-by-Point Response to Reviewers' Comments

We thank the reviewers for their thoughtful and constructive comments. To address the concerns raised in the comments, we have performed new experiments for more comprehensive evaluation of DeepETPicker. We have also expanded description of its technical details to provide further information. In addition, we have revised the manuscript to improve its clarity. All changes to the manuscript have been marked in red. We address specific concerns of the reviewers below.

Reviewer #1

Comments: The authors introduce a simple, but very effective, algorithm to identify macromolecules in tomograms. This is still an open problem in which any new contribution is welcome and expected to have an important impact on daily practice. The algorithm is technically sound and well-described. The authors present a very extensive analysis of its performance on various datasets, both simulated and experimental.

This reviewer finds that the manuscript can be published unaltered.

Response: We thank the reviewer for the positive and constructive comments. We perform the study in the hope of providing a fast and accurate tool to support automated 3D particle picking for high-resolution cryo-electron tomography *in situ*.

Reviewer #2

Comments: This manuscript introduced DeepETPicker, a novel deep-learning-based method for performing particle picking in 3D cryo-electron tomograms along with a user-friendly GUI. This method only requires the particle center locations to be provided for training purposes and does not require the voxel level segmentation maps of tomograms, unlike the other cryo-ET particle picking methods. The method is based on U-Net architecture like similar methods but incorporated several other modules including mask formation, MP-NMS, OT, etc. The authors validated their approach on two simulated benchmark datasets and two experimental ribosome tomogram datasets. DeepETPicker resulted in overall better performance in picking than the baseline and produced a higher resolution structure when combined with subtomogram averaging. By requiring only particle centers for training and by providing an interactive GUI for selecting particle centers, DeepETPicker stands out from other methods as a practically useful particle picking approach.

Despite showing much promise, there are several major and some minor concerns regarding the paper which are mentioned below

Response: We thank the reviewer for the positive and constructive comments. We address specific concerns of the reviewer below.

Comments: Major concerns:

1. The authors mentioned tuning hyperparameters as a drawback of reference-free DoG picking. However, their method requires several hyperparameters (t_g , t_{seg} , t_{dist}) as well. For t_g , they have always used 0.368. But is there any rationale behind choosing this value? The paper did not mention clearly how t_{seg} was chosen. The paper would benefit from discussing how the choice of hyperparameter values affects the final picking results.

Response: A major drawback of traditional particle picking methods such as the DoG is that their hyperparameters require frequent tuning for different datasets to achieve optimized performance. Even after hyperparameter tuning, their performance is generally inferior compared to that of deep learning models. Hyperparameters of deep neural network models also require tuning. But it is performed primarily in model training. We have added **Section A.3 to the Supplementary Methods** to provide a detailed discussion on the setting of hyperparameters for DeepETPicker. We have also modified the text accordingly (page 4, first paragraph, last sentence; page 7, last paragraph, line 2 from the bottom). In the following, for the reviewer's reference, we copy the discussion on the setting of t_g and t_{seg} from Supplementary Methods A.3.

The hyperparameter t_g determines the shape of Gaussian masks ($mask_{gaussian}$). The Gaussian mask $mask_{gaussian}$ becomes a ball mask when $t_g \geq$

$\exp(-0.5) \approx 0.607$, and it becomes a cubic mask when $t_g \leq \exp(-1.5) \approx 0.223$. Therefore, $mask_{gaussian}$ will be a Gaussian mask when $t_g \in (0.223, 0.607)$. To generate a Gaussian mask that is sufficiently different from ball/cubic masks, we choose the middle point between -0.5 and -1.5 and set $t_g = \exp(-1) \approx 0.368$ in our study. Related definitions have been revised for more clarity. See equations (1), (2), (3) and (4).

The output score maps of 3D-ResUNet are in the range of $[0,1]$, in which the value of each voxel denotes its probability score of belonging to a certain class. t_{seg} is a selected threshold that transforms a score maps into a binary map: a voxel with a value below t_{seg} is labelled as 0 and otherwise as 1 so that a binary map is generated. The influence of t_{seg} on the classification performance of DeepETPicker trained by different types of masks on SHREC2021 dataset is summarized in **Supplementary Table A1** of the revised Supplementary Information. The results show that the selecting of t_{seg} has little effect on the classification performance when it is within the range between 0.1 and 0.9. Therefore, we set the default value of t_{seg} to 0.5 in our study.

Please see **Supplementary Methods A.3** for a more detailed and comprehensive discussion on the setting of hyperparameters.

Comments: For the evaluation of simulated SHREC cryo-ET tomograms, the proposed method outperformed the baseline methods in several metrics, such as Precision, Recall, and F1 score. The improvement ranged between 2%-9%. However, in terms of another commonly used metric, AD (average distance), its performance was 24.3% less than that of the baseline method. Is there any justification behind this?

Response: For metrics such as Precision, Recall, and F1-score, the higher their values are the better. However, for AD (the average distance calculated in voxels from the center of the predicted particle to the ground truth), the lower its value is the better. A lower AD means the center of the predicted particle is closer to the ground truth. On simulated SHREC cryo-ET tomograms, DeepETPicker outperformed the baseline methods in metrics (precision, recall, F1-score) by a margin between 2%-9%. In terms of AD, DeepETPicker has an improvement of 24.3% over the baseline method.

Comments: One of the lucrative selling points of the paper is that it only requires labeling the particle center coordinates to prepare training data. The authors claimed that labeling the centers of the particles is easy and efficient, which was certainly the case for the tomograms they used for validation. But will it be the case for other tomograms as well, particularly when the particle size is small and the SNR of tomograms is lower? It would be nice to see a discussion on the ease of

selecting particle centers with respect to the particle size and the noise level in the tomogram.

Response: We appreciate the thoughtful and constructive comments. We added Gaussian noise of different levels to the SHREC2021 dataset and examined the influence of the noise level on performance of DeepETPicker in picking particles of different sizes (see **Fig. 2h**; **See also Supplementary Table 7** for detailed SNR levels). Under lower SNR levels, performance of DeepETPicker measured in F1-score gets worse (**Fig. 2h**). Moreover, the worsening in performance is more pronounced on smaller particles.

Under the typically low SNRs of tomograms, manually labelled particle centers generally deviate from real particle centers. But we find that the deviation is mostly smaller than half the particle radius. Taking EMPIAR-10499 as an example, we calculate the Euclidean distances between manually labeled centers and actual centers before and after refinement (**Supplementary Fig. 6**). We found that 80% of manually labelled centers are within 0.52 times the particle radius from the actual centers, and 90% of manually labelled centers are within 0.625 times the particle radius from the actual center. To further study the impact of the deviation of manual labeling on particle picking performance of DeepETPicker, we randomly add a shift within $0.5r$ and $0.7r$ to particle centers. We find that the randomly added shifts have minimal impact on the picking performance of DeepETPicker for all complexes with different sizes (**Fig. 2i**). Overall, this indicates that DeepETPicker has good robustness against the unavoidable deviation in particle center positions identified by manual labeling.

Comments: It is not clear how the method benefited from the use of coordinated convolution and image pyramid inputs. A detailed ablation study would be helpful.

Response: Coordinated convolution incorporates the spatial context of the input image into the convolutional filters, while image pyramid inputs effectively preserve features of the original input image at different resolution levels. These architectural design strategies effectively improve the performance of convolutional neural networks. We have revised the text to reflect these clarifications (see page 8, paragraph 3, last sentence).

To further verify the effectiveness of these strategies, we have also performed an ablation study on coordinated convolution and image pyramid inputs (**Supplementary Table A2**). We observe that coordinated convolution or image pyramid inputs can improve the mean F1-score by $\sim 1.5\%$ individually. Specifically, they improve the classification performance of tiny particles. When coordinated convolution and image pyramid inputs are utilized jointly, the mean F1-score of

classification on all complexes improves by 4.2%, and the mean F1-score of classification on tiny complexes improves by 8.1%.

Comments: The authors referred to supplementary figures 2c and 2d from the main manuscript. If supplementary figure 2 is very important to understand the manuscript, then I would recommend moving it to the main manuscript.

Response: We thank the reviewer for the suggestion and have moved the original **Supplementary Figures 2c and 2d** to **Figure 1** in the revised manuscript.

Comments: A table corresponding to Figure 2f would better help the reader to understand the relative improvement by DeepETPicker for simulated tomograms.

Response: We thank the reviewer for the suggestion and have added **Supplementary Table 5** to summarize relative improvement by DeepETPicker for simulated tomograms.

Comments: Can the method lead to any significant biological insight? Apparently, it does seem to directly present any new biological insight.

Response: We fully expect that DeepETPicker will enable significant biological insights by providing a fast and accurate tool for automated 3D particle picking for high-resolution cryo-electron tomography *in situ*. However, we feel that in this paper we should focus on the development and evaluation of DeepETPicker. To this end, we focus on presenting performance evaluation results on two simulated tomogram datasets and four experimental tomogram datasets. In our follow-up studies, we are using DeepETPicker to solve structural biology problems and to gain new biological insights.

Reviewer #3

Comments: This manuscript describes a supervised deep learning method for identifying macromolecule species in cryo-ET images. Evaluation is performed on a simulated benchmark dataset, as well as two experimental datasets. Comparison to two state-of-the-art methods (template matching and DeepFinder) is provided.

I recognize that the authors have put considerable efforts into the evaluation. Also, the method shows encouraging scores on the simulated benchmark. However, I consider that additional efforts are necessary for publication in Nature Communications. A major revision is needed. Below are the main reasons:

Response: We thank the reviewer for the positive and constructive comments. We address specific concerns of the reviewer below.

Comments: I notice that evaluation on experimental focus on ribosomes, which among the macromolecule species is the easiest to identify. While these results validate the ability of DeepETPicker to identify ribosomes, I consider that the evaluation should also be focused on macromolecules that are representative of contemporary research in structural biology, i.e. macromolecules with a molecular weight < 1MDa.

Response: We appreciate the suggestion. We have added results on another public cryo-ET dataset of *H. neapolitanus* alpha-carboxysomes (EMPIAR-11125) to evaluate the performance of DeepETPicker for picking smaller particles *in situ*. See page 24-25 of the revised manuscript for a summary of the results.

The molecular weight of alpha-carboxysome is 562 kDa. Following the same protocol, we compared performance of DeepETPicker, DeepFinder, crYOLO, and TM on this dataset. We found that DeepETPicker can pick true positive particles that are missed by DeepFinder, crYOLO and TM (**Fig. 5a**). Although crYOLO also picks particles not selected by DeepETPicker, these particles do not appear to be true positives upon initial visual inspection.

To check whether this observation is true, a comparison of particle detections between different methods (DeepETPicker, crYOLO, DeepFinder and TM) and manual annotation is carried out using precision and recall as metrics. At a fixed recall, DeepETPicker achieves the highest precision, followed by DeepFinder, TM and crYOLO (**Fig. 5b**). This indicates that DeepETPicker achieves the highest consistency with manual annotation. It also achieves the highest recall, indicating that more manual labelled particles are picked by DeepETPicker.

In addition, we performed sub-tomogram averaging to further check the quality of the picked particles (**Figs. 5c-d**) by utilizing the same analysis protocol for EMPIAR-10045. The global resolutions of reconstructed maps from particles picked by DeepETPicker, DeepFinder and TM are similar, at around 7 Å (**Fig. 5c**). However,

the particles picked by crYOLO fail to yield a correct reconstruction. In addition, the map reconstructed from particles picked by DeepETPicker shows more structural details and better local resolutions (**Fig. 5d**).

Comments: While this method presents novelties, I find that the method is an incremental work compared to DeepFinder. This is not something bad in itself, but it should be mentioned (please refer to [1] for more details). Some aspects of the method are unclear.

[1] Similarities and differences to DeepFinder

[1.1] What is similar to DeepFinder:

- use of a 3D semantic segmentation network to detect macromolecules
- same sampling strategy: sampling of 3D patches (subtomograms) centered on annotated coordinates, so that each patch contains at least 1 macromolecule. This is to mitigate the under-representation of the object classes w.r.t. to background class. This process is key for a successful training.
- use of "weak labels", i.e. placing spheres at annotated macromolecule positions
- overlap-tile strategy in inference stage
- use of Dice loss

[1.2] What is different:

- the network architecture.
- post-processing of the produced segmentation maps to obtain macromolecule coordinates. DeepFinder uses spatial mean-shift clustering, and DeepETPicker uses mean-pooling and non-maximum suppression.

The way you applied NMS here is unclear to me (even with suppl Fig2. D). Could you please elaborate? Also, see [4.2] for a remark on how you implemented NMS.

Response: We certainly acknowledge that DeepETPicker is inspired by the work of DeepFinder. We thank the reviewer for the thoughtful summary of the similarities and differences. In the following, we would also like to note some specific research contributions of our study.

a) A detailed study is conducted on different types of “weak labels” with different radius configurations.

For “weak labels”, DeepFinder place spheres (referred to as Ball masks in our study) with size-dependent radius at annotated macromolecule positions. In our study, we conduct a detailed study on different types of “weak labels” (Gaussian mask, Ball mask and Cubic mask) and different diameter settings (constant diameter, size-dependent diameter). Experimental results on simulated tomograms show that Gaussian masks provide stabler and better performance than Cubic masks and Ball

masks. Interestingly, we find that DeepETPicker trained by simplified masks with a constant diameter achieves essentially the same localization and classification performance as different sized particles. Therefore, different from DeepFinder, DeepETPicker uses Gaussian masks with constant diameters.

b) DeepETPicker is not sensitive to selection of loss functions.

Using simplified masks with constant diameters as the training labels effectively eliminates the problem of class imbalance and simplifies our selection of loss functions. We performed an ablation study for loss function, finding that different losses such as Dice loss, Focal loss, MSE loss and IoU loss give similar picking performances (see **Supplementary Table A4**).

c) A different implementation is developed for the overlap-tile strategy in the inference stage.

For the overlap-tile strategy, both DeepFinder and DeepETPicker crop the edge voxels of sub-tomogram to eliminate its poor segmentation accuracy, as shown in Figure R1a below. However, during the stage of merging subtomograms into a whole tomogram, patches of DeepFinder have an overlap area (Figure R1b) while patches of DeepETPicker have no overlap area (Figure R1c). Our experiment shows that overlap of cropped patches yields no improvement in performance. Further details can be found in the source code.

Figure R1. Comparison of overlap-tile strategies of Deepfinder and DeepETPicker

d) Several design features are incorporated in the architecture of DeepETPicker.

DeepETPicker uses 3D-ResUNet as its segmentation model. Specifically, the idea of residual connections in 2D-ResNet is incorporated into 3D-Unet, with the aim to better extract features from tomograms. ELU is used as the activation function to accelerate convergence of training. Incorporation of coordinated convolution and image pyramid inputs into 3D-ResUNet improves localization of particles.

e) Different post-processing methods

Compared to Deepfinder, DeepETPicker uses different post-processing to obtain macromolecule coordinates. Specifically, the Mean Pooling and Non-Maximum Suppression (MP-NMS) operation is proposed. Compared to clustering algorithms such as the mean-shift used in DeepFinder, the MP-NMS operation is substantially faster when performed using a GPU.

Regarding how NMS is applied.

MP-NMS operation is performed on 3D segmentation maps. To facilitate understanding of this operation, **Fig. 1h** in the revised manuscript shows an example of MP-NMS operation on 2D images. Further details regarding implementation of NMS can be found in the response to Comments [4.2] below (page 18, last paragraph).

Comments: Also, the way the authors implemented one of their method features, the "gaussian mask" is quite odd, and I have a number of doubts (see [2]).

[2] About the use of "weak labels"

[2.1] Implementation of "gaussian masks" The authors compare 3 ways of generating weak labels: cubic, ball (i.e. sphere) and gaussian. The benefits of using a gaussian function is that it produces smooth values (decreasing from the center), which can be useful for generating score-maps to be used as training targets (which would need a regression loss function, e.g. MSE). But the authors binarize the gaussian function so that the training targets correspond to segmentation maps (and proceed to use Dice loss). But a binarised 3D gaussian (as described in Eq. (3) of the manuscript) is... a sphere. Even though in Suppl. Fig. 2 (a) the masks "Bal-M" and "Gau-M" appear to be different, actually "Gau-M" is still a sphere whose radius is larger than the used observation window. So the authors present experiments to demonstrate that their gaussian masks are better than spherical masks (the latter being already used in cryoET, see ref [13] of the manuscript), but it turns out both are identical. Could you please explain?

Response: We regret the confusion caused by the text. Indeed, a similar concern was also raised by Reviewer #2. The question of the reviewer can be answered by looking at the setting of hyperparameter t_g . Please see our response above to the first question of Reviewer #2, specifically the text regarding the setting of hyperparameter t_g . For the reviewer's reference, we copy the answer below. We have also revised related definitions for clarity. See equations (1), (2), (3) and (4) on page 7.

“The hyperparameter t_g determines the shape of Gaussian masks ($mask_{gaussian}$). The Gaussian mask $mask_{gaussian}$ becomes a ball mask when $t_g \geq \exp(-0.5) \approx 0.607$, and it becomes a cubic mask when $t_g \leq \exp(-1.5) \approx 0.223$. Therefore, $mask_{gaussian}$ will be a Gaussian mask when $t_g \in (0.223, 0.607)$. To generate a Gaussian mask that is sufficiently different from ball/cubic masks, we choose the middle point between -0.5 and -1.5 and set $t_g = \exp(-1) \approx 0.368$ in our study. Related definitions have been revised for more clarity. See equations (1), (2), (3) and (4).”

Comments: [2.2] Interrogation about the use of spherical masks

You advocate for the use spherical/"gaussian" masks, which is a good approximation for macromolecules whose shape is sphere-like. How do you plan to handle the case of elongated macromolecules (for ex. double-capped 26S proteasome, EMD-3932)? In this case the overlap between the real shape (cylindrical) and the modelled shape (spherical) is reduced, and hence the label noise is increased, which may prevent the training from converging.

Response: For particles with different shapes and sizes, we first experimented on the dataset from the SHREC2021 challenge. The dataset contains 12 types of particles whose molecular weights range from 70KDa to 3325.59 kDa and whose shapes vary from spherical and elongated. DeepETPicker achieved the best overall picking performance for this dataset using simplified Gaussian masks with a constant diameter.

Unfortunately, we could not find the raw tilt-series for the double-capped 26S proteasome (EMD-3932). Instead, to evaluate the ability of DeepETPicker to pick elongated macromolecules, we chose the public cryo-ET dataset of T20S proteasome (accession code EMPIAR-10651), which has a cylindrical shape. We picked particles using DeepETPicker, crYOLO, DeepFinder, TM methods as well as by manual annotation. The results showed that DeepETPicker can pick true positive particles that are missed by crYOLO and DeepFinder (**Supplementary Fig. 10a**). Regarding precision and recall, DeepETPicker and TM achieve comparable performance, which are slightly better than that of DeepFinder and much better than that of crYOLO (**Supplementary Fig. 10b**). The map reconstructed from particles picked by DeepETPicker shows more structural details and better local resolutions (**Supplementary Fig. 10d**). See page 22 of the revised manuscript for further details.

Comments: [2.3] Remark on how the "weak label" strategy is presented
In the abstract, the authors write: "Training of DeepETPicker requires only weak supervision by simplified Gaussian-type labels, which reduce the burden of manual annotation of tomograms under very low signal-to-noise ratios"
Also, on p.5, l.12: "Model training of DeepETPicker requires only weak supervision using simplified labels, resulting in substantially reduced cost of manual annotation".

The authors make it seem as if their "simplified labels" will save biologists a lot of time in the annotation process, who would otherwise have to produce voxel-wise annotations. In reality, producing manually voxel-wise annotations for macromolecules is an impossible task, because of very weak contrast and blurry edges. So in fact, without the "weak annotation strategy", their method would simply be impracticable, which is quite different to the authors statement.

Response: Compared to existing deep-learning based methods such as DeepFinder, DeepETPicker requires less training data to achieve the same level of performance (Fig. 2g). This reduces the burden of manual annotation of tomograms under very low signal-to-noise ratios. For example, for EMPIAR-10499, DeepETPicker requires 117 particles for training to achieve relatively good performance while DeepFinder requires substantially more.

We have revised the sentence "Model training of DeepETPicker requires only weak supervision using simplified labels, resulting in substantially reduced cost of manual annotation" in the abstract as "The training of DeepETPicker requires only weak supervision with low numbers of simplified Gaussian-type labels, reducing the burden of manual annotation of tomograms under very low signal-to-noise ratios." (page 2, line 9 of the revised manuscript).

We have revised the sentences "Training of DeepETPicker requires only weak supervision by simplified Gaussian-type labels, which reduce the burden of manual annotation of tomograms under very low signal-to-noise ratios" as "The model training process of DeepETPicker requires only weak supervision using simplified labels and fewer training labels to attain performance comparable to that of competing methods, which reduce the cost of manual annotation." (page 5, line 11 of the revised manuscript).

Comments: I have several concerns about the evaluation protocol on the experimental datasets (see [3]).

[3] About the evaluation protocol for the EMPIAR datasets

[3.1] Concern about the described subtomogram averaging pipeline

According to p.13 l.28, the authors use 3D classification in their subtomogram averaging procedure, which is very disturbing. This means that the authors apply post-classification to the particle picking method outputs. They should compare

the raw output of the methods, else they do not measure the method performances, but the performance of a [method + post-classification] pipeline. This basically invalidates all subtomogram averaging results on EMPIAR-10045 and EMPIAR-10499.

Response: For subtomogram averaging of each dataset, we used the same data processing process to ensure fair comparison of different algorithms (Supplementary Fig. 4). No particles were removed in post-processing, avoiding influence of subjective biases. On page 17, line 23-26 of the original manuscript, we wrote “To make an objective comparison, we performed no particle screening during the subsequent alignment and classification because otherwise the quality measurement of picked particles will be affected by the screening protocols used. We only set one class in the 3D classification step...”. Finally, all particles are used for 3D refinement by *Relion* to compare the final resolution.

Although we performed 3D classification, we used only one class for the purpose of calculating metrics such as log-likelihood contribution and max value probability, which could only be obtained from 3D classification. We have revised the text to note that only one class is used in 3D classification. See page 15 line 2 and line 19 as well as page 16 line 15.

In the revised manuscript, we have added the comparison of raw output for all methods. Specifically, before post-processing, we compared the precision/recall of the 4 tested methods by comparing the coordinates of particles picked by them with the coordinates of particles manually picked by experts (Figs. 3-5).

Comments: [3.2] Questions about your evaluation criteria

The term "quality of particles" is mentioned around 15 times in the text, but this term is vague and the authors do not define what they mean by that. For all experiments involving subtomogram averaging, they put a considerable amount of effort measuring this "quality", see for example:

p.10, l.23: "For real experimental datasets without ground truth, we used B-factor, global resolution, local resolution, and log-likelihood distribution to evaluate and compare the quality of particles picked by DeepETPicker and other competing state-of-the-art methods".

What exactly do you mean by particle "quality"? The SNR of the subtomogram? How does measuring this "quality" help quantifying the performance of a particle picker? Shouldn't a good particle picker detect all target macromolecules, regardless of the SNR amount (even the most noisiest ones)?

As a reminder, the resolution of a subtomogram average quantifies the structural homogeneity of picked particles. The resolution is low when the average is noisy

(e.g. particle count is low), or when the average is blurry (e.g. particles are not similar enough). Structural heterogeneity occurs when: (i) picked particles are not of the same macromolecule species (i.e. false positives), and (ii) picked particles are of the same macromolecule species, but with different structural conformations.

Here is an example to illustrate my point. The 26S proteasome has 2 assembly states: single-caped, double-caped. Including both states in the average will produce blur at the cap locations (i.e. resolution is low at these locations). As for the detector, it did not make any mistakes, as it picked proteasomes correctly (regardless of their assembly state). So in this case, the resolution of obtained subtomogram average does not reflect well the performance of the particle picker.

I think that it would be beneficial to the paper to discuss how the resolution of an obtained subtomogram average is correlated to the performance of a particle picker (and under which conditions).

Response: We appreciate the thoughtful comments. In the Methods section, we have added a subsection titled “Quality metrics for picked particles” to summarize the specific metrics used in our study. In addition to precision, recall, and F1-score, for real experimental datasets without ground truths, we use the B-factor, global resolution, local resolution, and log-likelihood distribution to evaluate and compare the quality of the particles picked by DeepETPicker and other competing methods. Conceptually, particle “quality” can be assessed by the authenticity of the particles, the accuracy of the particle coordinates, and the contribution of the particles to subtomogram averaging.

In our paper, different methods (DeepETPicker, crYOLO, DeepFinder and TM) are compared on the same dataset following the same analysis protocol. In this case, because other conditions are kept the same, if particles picked by a certain method achieves higher resolution than competing methods, we consider the particles picked by this method to be of higher overall quality. We have added this discussion on page 12, paragraph 2, last sentence.

Comments: [3.3] When comparing DeepETPicker to template matching and DeepFinder on the EMPIAR datasets, the authors do not explain how they chose the score thresholds of the last two methods. Yet the performance of TM and DeepFinder depends greatly on the chosen threshold, and therefore it should be mentioned when making a comparison.

Response: We have added Supplementary Methods Section A.7 to explain how these thresholds are set in our study. For the reviewer’s reference, the text below is copied from that section.

For DeepFinder, it would generate a file with five columns, i.e., class label, x, y, z and cluster size. For the EMPIAR-10045 and EMPIAR-10499 datasets, the diameter of ribosomes is about 23~24 voxels. During the training stage, we use spheres with a radius of 11 as the labels. For a sphere with a diameter of 23, its volume can be calculated as $V = \frac{4\pi r^3}{3} = \frac{4\pi \times 11.5^3}{3}$. In the inference stage, particles with a cluster size in the range of [0.1V, 2V] are selected as the final result for EMPIAR-10045. Particles with a cluster size in the range of [0.2V, 2V] are selected as the final result for EMPIAR-10499. For EMPIAR-10651, we use spheres with a radius of 11 as the labels, and its volume can be calculated as $V = \frac{4\pi r^3}{3} = \frac{4\pi \times 11^3}{3}$. In the inference stage, particles with a cluster size in the range of [0.1V, 2V] are selected as the final result for EMPIAR-10651. For EMPIAR-11125, we use spheres with a radius of 7 as the labels, and its volume can be calculated as $V = \frac{4\pi r^3}{3} = \frac{4\pi \times 7^3}{3}$. In the inference stage, particles with a cluster size in the range of [0.1V, 4V] are selected as the final result for EMPIAR-11125.

For template matching, we use mainly the template matching function “dynamo_match” of Dynamo. There are two parameters that may affect particle selection. The parameter 'cr' (cone range) defines orientations that will be looked for inside a cone. In our experiment, we use the most typical value of 360 (sample the full sphere). The parameter 'cs' (cone sampling) determines the scanning density inside the sphere. In our experiment, we use the most typical values of 30 (sample the full sphere). It will generate tbl-format table files, where the tenth column shows the cross-correction coefficient. For each tomogram, we obtain a plot of the cross-correlation values found on the local maxima of the cc volume with the order. The cross-correlation values of the peaks appeared in an ascending order. We check the quality of the peaks by auxiliary clicking on the curve to select one particle and then selecting some visualization option. We click on a couple of particles in the area of kink in the cross-correlation to roughly estimate the cross-correlation threshold. The detailed thresholds for TM for different datasets are provided in **Supplementary Table A6**.

Comments: [3.4] For all subtomogram averages you show, please indicate the number of particles used to compute it, as poor resolution may be due to low particle count. I noticed that you give some particle counts in Suppl. Table 9, however it would be easier to interpret the averages if you also display these values on the figures.

Response: We thank the reviewer for the suggestion. For all subtomogram averages (FSC curves and subtomogram averaging maps), we have shown the number of particles used in their computation. More details can be found in **Figs. 3-5**,

Supplementary Figs. 8-9, and Supplementary Figs. 12-13 of the revised manuscript.

Comments: [3.5] According to Suppl. Table 9, we have:

- Subset $TM \cap \text{DeepETPicker}$: nb_of_particles=3662, resolution=19.2 Å

- Subset $TM - \text{DeepETPicker}$: nb_of_particles=6333, resolution=19.2 Å

Both subset averages have the same resolution of 19.2Å, which suggests that both subsets have a good true positive rate. This also suggests that DeepETPicker missed 6333 ribosomes, which is a lot. Could you please comment on this?

Response: Resolution represents the consistency of two half sets: even if both half sets contain error information, it is possible to obtain high resolution. Therefore, although subsets ‘ $TM \cap \text{DeepETPicker}$ ’ and ‘ $TM - \text{DeepETPicker}$ ’ have the same resolution of 19.2 Å, it does not necessarily mean that they have a high true positive rate. This is confirmed by the reconstruction maps of subtomogram averaging for subsets of ‘ $TM \cap \text{DeepETPicker}$ ’ and ‘ $TM - \text{DeepETPicker}$ ’ (**Supplementary Fig. 12**). The subset ‘ $TM - \text{DeepETPicker}$ ’ yields erroneous reconstruction maps, indicating that the particles picked by TM by not DeepETPicker are mostly false positives with poor quality.

Comments: [3.6] For the intersection of two sets A and B, we should have $A \cap B = B \cap A$. Could you please explain why this is not the case in Suppl. Tables 8 and 9? For ex. in Suppl. Table 9, $\text{DeepETPicker} \cap TM$, and $TM \cap \text{DeepETPicker}$ contain 3618 and 3662 particles, respectively (i.e. a difference of 44 particles!).

Response: Mathematically, we would expect $A \cap B = B \cap A$. However, in our study, if the minimal Euclidean distance between two particles is lower than a specific threshold t_{dist} , which normally is set to be half of the diameter of the particle, the two particles are considered the same. An illustration of the intersection and difference sets of particles picked by two methods A and B is given in **Supplementary Fig. 3**. In **Case i**, Method A identifies a particle as a single particle, but method B identifies the same particle as two separate particles with different centers. In **Case ii**, Method A identifies a particle as two separate particles with different centers, but method B identifies it as a single particle. Either case i or ii may lead to different number of intersection particles of two methods, indicating that the number of intersection particles of two methods may not be exactly the same. Specifically, taking Case i as an example, under the above definition of “same particles”, the particle will count as 1 in $A \cap B$ and will count as 2 in $B \cap A$. In summary, the seemingly counterintuitive asymmetry is caused the above definition of “same particles”. Although this counterintuitive asymmetry can be eliminated by modifying the definition of “same particles”, we choose not to do so because the current way of computing $A \cap B$ and $B \cap A$ provides useful information. We have added this discussion to the legend of Supplementary Fig. 3.

Comments: [3.7] Why don't you compare the detections to the manual annotations (via precision, recall, F1-score)? It is true that manual annotations are not a ground truth, but it gives an indicator of how close the particle picking method is to what an expert has picked.

Response: The comparison has been performed. Expert annotation of EMPIAR-10045 is provided by the original paper. We have obtained expert manual annotations for EMPIAR-10499, EMPIAR-10651 and EMPIAR-11125. Results of comparing particle detection results of different methods (DeepETPicker, crYOLO, DeepFinder and Template Matching) and manual annotation are shown in **Figs. 3-5** and **Supplementary Fig. 10** of the revised manuscript.

Comments: [3.8] How do you explain that DeepFinder performs well on EMPIAR-10045 and poorly on EMPIAR-10499? Given that DeepFinder has been shown to have state-of-the-art performance for detecting ribosomes (and even differentiate between two binding states of the ribosome) in cellular tomograms (*Chlamydomonas reinhardtii* cells)?

Response: EMPIAR-10045 is a dataset of purified ribosomes, while EMPIAR-10499 is a dataset of ribosomes *in situ*. Compared to EMPIAR-10045, EMPIAR-10499 has a lower signal-to-noise ratio (SNR) and more complex background, making it more difficult to pick particles from EMPIAR-10499. In the original paper of DeepFinder, the dataset used for training DeepFinder composed of multiple tomograms and has thousands of particles (e.g. the training dataset of *Chlamydomonas reinhardtii* cells consists of 6834 mb-robos and 6687 ct-ribo particles). However, for both EMPIAR-10045 and EMPIAR-10499 datasets, our study uses only one annotated tomogram for training, which contains only hundreds of particles. According to **Fig. 2g** for the SHREC2020 simulated datasets, DeepFinder underperforms DeepETPicker when the number of training tomograms is low.

Comments: [3.9] p.13, l.22: "We selected 150 particles from manual annotation to train DeepETPicker, DeepFinder and TM." Did you make sure that the train and test particles do not originate from the same tomograms? When sampling 3D patches centered around annotated particles, there are good chances that the patches also includes additional neighboring particles. If the split into train and test sets is not done carefully, there is a risk of overlap between train and test sets (i.e. a test patch may contain particles from train set), which results in metrics being overestimated.

Response: We appreciate the thoughtful comments. We have added **Section A.8** to the **Supplementary Methods** to provide details on the split of

training/validation/test sets. For all the four experimental tomogram datasets, the coordinates of the manually picked particles are sorted in the order of z, y, and x from the smallest to the largest before split of the training and validation sets. This sorting strategy reduces the risk of overlap between the training and the validation sets.

For comparing performance of DeepETPicker with other methods on the simulated tomogram datasets, we follow the protocol provided by the SHREC challenges. For comparison of performance of DeepETPicker with competing methods on the real tomogram datasets, we use all the tomograms as test datasets for all the methods. For EMPIAR-10045 and EMPIAR-10499, to avoid the performance metrics being overestimated, we remove training particles from the detected particles for all methods.

For EMPIAR-10651 and EMPIAR-11125, we can only find 3 raw tilt-series. Consequently, the total number of particles detected by all methods are quite low. To ensure there are enough particles for subtomogram averaging, we choose not to remove the training particles. We agree with the reviewer and acknowledge that the performance metrics of all methods on these two datasets are overestimated. We have compared the results with and without a split of the training and test sets in **Supplementary Figure A1**. The conclusions on comparing performance of different methods do not change qualitatively.

Comments: [3.10] p.14, l.13: "We manually picked 117 particles to train crYOLO and DeepETPicker, and 703 particles to train DeepFinder" Why are you using more particles for DeepFinder? Does this mean that the train and test sets are not the same for all compared methods? Please describe clearly the train, valid and test sets for the EMPIAR-100045 and EMPIAR-10499 datasets.

Response: The split of training/validation/test sets are described in detail in Section A.8 of the Supplementary Methods. For EMPIAR-10499, a total of 117 particles are manually labelled. For crYOLO and DeepETPicker, 106 particles are used for training. However, because using 106 particles to train DeepFinder failed, we increased the training particle number to 650 particles.

For the reviewer's reference, the following text is copied from Section A.8 of the Supplementary Methods.

For EMPIAR-10045, 150 particles were manually labelled along consecutive different z slices. The coordinates of these 150 particles are sorted in order of z, y, and x from the smallest to the largest. For crYOLO, DeepFinder and DeepETPicker,

the first 135 sorted particles are used for training, and the remaining 15 particles are used for validation. In the inference stage, the trained model of each method is used to pick particles on all tomograms.

For EMPIAR-10499, 117 particles were manually labelled along consecutive different z slices. The coordinates of these 117 particles are sorted in order of z, y, and x from the smallest to the largest. For crYOLO and DeepETPicker, the first 106 particles are used for training and the remaining 11 particles are used for validation. For DeepFinder, because its initial training using 106 particles fails to converge, we increase the number of manually labelled particles to 703, with the first 650 particles used for training and the remaining 53 particles used for validation. In the inference stage, the trained model of each method is used to pick particles on all tomograms.

Comments: [4] Other remarks

[4.0] The authors announce their software to be open-source, but the github page they link to does not contain the source code. However it does contain a docker image, which I was not able to execute (this might be due to my configuration). I could nevertheless access their code through the docker image, but it would be better to be able to visualize the code in github directly. This would allow users to estimate the quality of the code more easily. For example, it seems like the authors repository does not follow the standard organisation of a python package (as needed for distribution on PyPI). Following these standards allow for better reproducibility and reusability of the method. I am well aware that this is not a condition for publishing, please consider this as a strong recommendation.

Response: We really appreciate the recommendation. We have initially made the source code publicly accessible through GitHub. Docker deployment is a popular method for distributing software. Multiple end users have provided feedback that the Docker image runs successfully. As recommended, we have now added another installation method using conda so that the code can be directly visualized in GitHub. Further details can be found in <https://github.com/cbmi-group/DeepETPicker>.

Comments: [4.1] p.8 l.11: "The value of each voxel in the segmentation maps [...] denotes its probability score of belonging to a certain class, and the score is in the range of [0,1]" You seem to confuse the terms "segmentation map" and "score map". A segmentation map has integer values in the range of [0, n_classes]. A score map has float values in the range of [0,1].

Response: We have revised the sentence as “**The value of each voxel in the score map generated by 3D-ResUNet denotes its probability of belonging to a certain class, which is in the range of [0, 1].**” See page 9, third paragraph, first sentence.

Comments: [4.2] p.8, l.17: "Then the proposed MP-NMS operation [...] are performed on the binary image to obtain local maxima" and then p.8, l.24: "The larger the local maximum, the higher the probability that this is a particle"

I find it quite odd to estimate local maxima on binary images. Even more so if you use this value for non-maximum suppression, given that all local maxima will have the same value: 1.

Response: We regret the confusion caused by our writing. We have now revised the sentence as “the proposed MP-NMS operation, consisting of multiple iterations of mean pooling (MP) and one iteration of nonmaximum suppression, is performed on the binary map as the initial input”. The binary map is used only as the initial input. Outputs of different iterations of mean pooling are no longer binary. The following is a more detailed explanation of the procedure.

To obtain macromolecule coordinates from the produced segmentation maps, we propose Mean Pooling and Non-Maximum Suppression (MP-NMS) operation. **Fig. 1h** shows an example of MP-NMS operation on 2D images. MP-NMS mainly consists of two steps. The first step of MP-NMS is to transform a binary mask to a soft mask by using multiple iterations of mean-pooling (MP), as shown in the first row in **Fig. 1h**. After each MP operation, voxels in mask edge are pulled closer to the voxel value of background. As the number of rounds of MP increases, all voxels of the mask will be changed. Eventually, the binary mask will be converted into a soft mask. The further a voxel in the mask is from the background, the larger its value. Thus, the centroid of the mask should be a local maximum. The second step is to get the local maximums (coordinate of particle centroid) by performing non-maximum suppression on the soft mask, as shown in the third row in **Fig. 1h**. Each local maximum can be considered as a candidate particle center.

Comments: [4.3] p.8, l.26: "Compared with clustering algorithm such as mean-shift used in DeepFinder, our proposed MP-NMS operation is substantially faster when accelerated using a GPU" Here you should point to Suppl. Table 6. Which brings me to the next point: this table is supposed to compare computing times, however for each method a different GPU is used (sometimes even multiple GPUs). Therefore these numbers are not comparable, and you should not call this table a "comparison". Also, the computing times reported in Fig.2 (e) do not mention which hardware was used.

Response: We thank the review for the constructive comments. Training and inference of DeepETPicker was performed on a single Nvidia GeForce GTX 2080Ti GPU. We have revised the text accordingly (page 6, second paragraph, last line; page 10, first paragraph, last line) and have updated Supplementary Table 6.

For deep learning methods, their inference time in practice is mainly determined by the GPU hardware used. Specifically, the inference time of a GPU is mainly determined by its performance parameter TFLOPs (i.e., the number of floating-point operations per second) and the number of GPU used. To compare the inference time of different methods calculated by different GPUs, we estimated the total number of floating-point operations of each method based on the GPU used, GPU number, and inference time. Then the total number of floating-point operations of different methods are normalized by that of DeepETPicker to obtain what we refer to as estimated speedup ratio, which estimates the acceleration effect of DeepETPicker (**Supplementary Table 6**). We found that compared with e.g., DeepFinder, DeepETPicker takes $\sim 1/21.76$ of its inference time. In our revised manuscript, we have changed “comparison of computing time” into “reported computing time” to make it more accurate.

Supplementary Table 6 | Reported computing time in training and inference stages for processing one SHREC2021 tomogram of size $200 \times 512 \times 512$. Calculation of estimated speedup ratios of DeepETPicker in the inference stage is based on the assumption that the calculation is mainly completed by the GPU.

Methods	Training stage	Inference stage	Hardware	FP32 TFLOPS	Estimated speedup ratio of DeepETPicker in inference stage
URFinder	300h	2h6m	Nvidia Quadro RTX 8000 GPU (2x)	16.3×2	654.42
DeepFinder	50h	20m	Nvidia M40 (1x)	6.83	21.76
U-CLSTM	120h	15m	Nvidia Quadro RTX-5000 GPU (1x)	11.2	26.77
MC DS Net	22h	5m	Nvidia GeForce RTX 3090 GPU (1x)	35.58	28.34
YOPO	8h	40m	Nvidia GeForce Titan X GPU (1x)	6.69	42.63
CFN	96h	-	Nvidia GeForce RTX 3090 GPU (2x)	35.58×2	-
TM-F/TM GPU	N/A	4h26m	Nvidia GeForce GTX 1080Ti (1x)	11.34	480.58
DeepETPicker	17h	28s	Nvidia GeForce GTX 2080Ti (1x)	13.45	1.00

Comments: [4.4] p.16, l.16: "DeepETPicker achieves similar performance improvements over methods in the SHREC2020 challenge. This is because DeepETPicker utilizes customized lightweight and efficient architecture in its 3D-ResUNet segmentation model as well as a GPU accelerated pooling-based postprocessing method"

The authors claim that their architecture is more lightweight than competing methods in the SHREC2020 challenge, and that this is one of the reasons why DeepETPicker performs better. However, when I compare their architecture (21 convolutional layers) on Suppl. Fig. 2 to the architecture (15 convolutional layers) used in DeepFinder (one of the competitors in SHREC2020), it seems to me that the latter has less parameters (taking into account the number and the size of the filters).

Response: It appears that the sentence causes some misunderstanding. We attribute the good performance of DeepETPicker to the synergy of several factors, including its lightweight architecture and its GPU-accelerated postprocessing. We have not made any direct comparison of the architecture of DeepETPicker with those of competing methods. To avoid the misunderstanding, we have revised the sentence as “The customized lightweight and efficient architecture of 3D-ResUNet as well as its GPU-accelerated pooling-based postprocessing method, namely MP-NMP, are key factors that contribute to the performance of DeepETPicker.” (page 18, second paragraph, last sentence).

Supplementary Fig. 2 in the original manuscript shows a specific architecture of 3D-ResUNet with the numbers of channels as $[c_1, c_2, c_3, c_4] = [24, 48, 72, 108]$. To check the architectural efficiency of 3D-ResUNet, we decrease the channels of feature maps to $[8, 16, 24, 36]$. The localization and classification performance are comparable to that of $[24, 48, 72, 108]$ and are better than that of DeepFinder (**Supplementary Table A3**). Furthermore, 3D-ResUNet with channels $[8, 16, 24, 36]$ has a model size of 3.4M, which is substantially smaller than DeepFinder with a model size of 11M, validating the efficient architecture of DeepETPicker. In our revised manuscript, we modified the **Supplementary Fig. 2** and added an ablation study on the number of channels of 3D-ResUNet to evaluate its architectural efficiency (**Supplementary Table A3**).

Comments: [4.5] p.9, l.23: "Specifically, the following transformations are performed on the training datasets: random cropping, mirror transformation, elastic deformation less than 5%, scaling in the range of $[0.95, 1.05]$, and random rotation in angles withing $[-15^\circ, 15^\circ]$ "

In my opinion, data augmentations should not change the data distribution in ways that are not desirable. Here is what I mean by that: firstly, the main clue for identifying a macromolecule species is its shape, which includes its chirality. Therefore, should you be using mirror operations? Secondly, cryo electron tomograms have an anisotropic resolution. If you use random rotations, the orientation of mentioned anisotropy is changed. Should your model be invariant to the anisotropy orientation?

Response: We thank the reviewer for the thoughtful comments. Indeed, biomacromolecules have chirality, and cryo-electron tomograms have anisotropic resolutions. The data augmentation used in our study i.e., mirror transformation and spatial transformation, are not realistic from a purely biological perspective. However, data augmentation operations such as rotation and mirror transformation increase the diversity of tomograms for a given dataset. These operations are standard practices in deep learning. Our ablation study showed that data augmentation operations such as mirror transformation and spatial transformation effectively improve the particle picking performance of DeepETPicker (**Supplementary Table A5**), although we currently cannot explain this improvement from a purely biological perspective. In addition, cryo-ET particles in experimental tomograms generally exhibit a preferred range of orientations. In principle, increasing the diversity of tomograms via random rotation can enhance the picking performance of the DeepETPicker for particles in other orientations, especially when the number of particles in a certain orientation is low. We plan to investigate these issues in depth in follow-up studies.

Comments: [4.6] p.15, l.23: "Precise localization of particle centers is crucial for sub-tomogram averaging" While it is certainly desirable to obtain a precise localization, it is not "crucial" for sub-tomogram averaging. Having small localisation errors is fine, as in subtomogram averaging, not only the macromolecule orientation is estimated, but also a spatial shift.

Response: We thank the reviewer for the thoughtful comment and agree that “crucial” is not a proper word here. In our revised manuscript, we have revised the term “crucial” to “important”. See page18, first paragraph, line 1.

We are aware that in subtomogram averaging, not only the macromolecule orientation but also a spatial shift is estimated. To a certain extent, the center coordinates of picked particles are allowed to deviate from the real particle center coordinates. Currently, however, when calculating the alignment, the alignment algorithm generally needs to set the search range. All orientational searches, or integrations in the statistical approach, are performed over the full six dimensions (i.e., three Euler angles and three translations). If the picked center is largely consistent with the real particle center, it is easy to detect correct orientation. But if the localization deviation is large and outside the set search range, it will not be possible to search and correct within the limited range. Larger search range can handle the case of larger localization errors, but as the search range increases, the amount of calculation increases substantially.

Specifically, we take “Scheres, S. H. (2012). A Bayesian view on cryo-EM structure determination. *Journal of molecular biology*, 415(2), 406-418” as a reference. This article proposes a three-dimensional reconstruction method based on the Bayesian framework, which can automatically correct the deviation of the coordinate center during particle selection, thereby improving the reconstruction resolution. It describes in detail how maximum a posteriori (MAP) estimation sets up and searches for three Euler angles and coordinate offsets when performing single particle analysis. The MAP algorithm in RELION searches iteratively. When the localization deviation is large, the initial search may be wrong, and then it will be searching in the wrong direction, which cannot be corrected.

Comments: [4.7] p.2 l.7: "adoption of automated particle-picking methods remains limited because of their limitations in [...] training cost" This is in my opinion an odd wording. How can a training cost be "limited"? With respect to what?

Response: We used the term “training cost” to refer to the large numbers of manual annotations often required for training the deep-learning models. Existing deep-learning models may require up to thousands of manually annotated particles for training, which are laborious and time-consuming to generate in practice. In our revised manuscript, we have changed the term “training cost” to “manual annotation cost” for more clarity. See page 2 line 7 in the abstract.

Comments: [4.8] p.4, l.9: when speaking of the DoG picker, you should mention that this picker is not class specific, unlike all other methods mentioned. DoG picks particles regardless of their class.

Response: We appreciate the suggestion and have revised the manuscript accordingly. See page 4, first paragraph, line 3 from the bottom.

Comments: [4.9] p.10, l.14-18: "TP is shorthand for 'true positive', namely a positive particle is predicted to be positive ..."

The authors provide detailed definitions for the terms true positives, false positives and false negatives. In my opinion this is not necessary, and it will make you save some lines.

Response: We appreciate the suggestion and have removed the detailed definitions. See page 11, first paragraph.

Comments: [4.10] p.19, l.3: "Therefore, although the additional particles by DeepFinder and TM improves the SNR of the whole dataset [...]"

This statement is bizarre. How does a particle picking method improve the SNR of the "whole dataset"? I assume that by "whole dataset" you mean the tomogram set. You are not performing denoising, so what do you mean by "improve the SNR"?

Response: We regret the confusion caused by our writing. The term "whole dataset" should be "half maps", i.e., reconstruction maps of the two independent halves of the datasets, for the FSC curves (see page 11 last paragraph of the Methods section). As the number of picked particles increases during the averaging process, the noise of the half maps will be suppressed, and the signal-to-noise ratio of the half maps will improve, but this does not guarantee the authenticity of the signal. In our revised manuscript, we have changed the term "whole dataset" to "half maps" for more clarity. See page 22, first paragraph, line 9.

Comments: [4.11] p.19, l.12: "The crowded cellular environment adds additional background to make particle picking [...] more difficult".

This sentence is clumsy. There is no background that is being added (the "amount" of background stays the same). With a crowded cellular environment, the background becomes rather more complex/challenging.

Response: We appreciate the suggestion. We have revised the sentence as "The crowded cellular environment poses a complex and challenging background for

particle localization and identification”. See page 23, first paragraph in the revised manuscript.

Reviewer #2 (Remarks to the Author):

The authors have improved the manuscript from the previous submission. They have properly addressed the reviewers' concerns on the difference between Gaussian masks and spherical masks, measuring quality with subtomogram averaging, the method's generalization ability to non-spherical structures, and applicability to structures smaller than ribosomes. The authors have addressed the minor remarks as well and made appropriate changes to the manuscript.

In my opinion, the proposed method, DeepETPicker, will be a very useful tool for the cryo-ET, as well as, the structural biology community to perform semi-automated picking of macromolecules in situ. Moreover, DeepETPicker can be used not only for picking large macromolecules like ribosome, but also for smaller non-spherical macromolecules. Consequently, I vote for accepting the manuscript unaltered in its current form.

Reviewer #3 (Remarks to the Author):

I would like to thank the authors for their work and their efforts in answering my questions. With these answers, new informations came into light, which confirm concerns I described in my first review. As a result, I am sorry to announce that I consider a major revision to be necessary. I have two main concerns:

1/ The train, validation and test sets are not clearly separated.

1.1/ During training, the procedure samples 3D patches from the tomogram set. I asked if the train and validation patches originate from different tomograms, because if not, there is a risk that a patch contains both train and validation particles (see my last review, comments 3.9 and 3.10). The authors answered:

"For all the four experimental tomogram datasets, the coordinates of the manually picked particles are sorted in the order of z, y, and x from the smallest to the largest before split of the training and validation sets. This sorting strategy reduces the risk of overlap between the training and the validation sets."

Maybe it reduces the risk, but it does not eliminate it. For a rigorous statistical analysis, no overlap between train, validation and test sets should be allowed.

1.2/ But most importantly, the authors added new information about their datasets:

"For EMPIAR-10045, 150 particles were manually labelled [...]. The coordinates of these 150 particles are sorted in order of z, y, and x from the smallest to the largest. For crYOLO, DeepFinder and DeepETPicker, the first 135 sorted particles are used for training, and the remaining 15 particles are used for validation. In the inference stage, the trained model of each method is used particles on all tomograms."

So first, the authors divide the dataset into training and validation sets. Then, for evaluating the method, they proceed to join the training and validation sets to form their test set, which should not be allowed. They proceed in a similar way for other datasets.

Answering to my previous concerns, the authors write that for some of the datasets: "to avoid the performance metrics being overestimated, we remove training particles from the detected particles for all methods."

So I conclude that for the dataset cited above (EMPIAR-10045), after removing the training particles, the test set is constituted of only 15 particles, which is clearly not enough to have a strong estimate of performance. And I also conclude that their test set is identical to their

validation set.

To justify their protocol, the authors wrote:

"We agree with the reviewer and acknowledge that the performance metrics of all methods on these two datasets are overestimated. We have compared the results with and without a split of the training and test sets in Supplementary Figure A1. The conclusions on comparing performance of different methods do not change qualitatively."

While I understand partially that the authors proceed in this way because the sizes of their datasets are limited, this is certainly not a standard protocol for evaluating a machine learning method.

It is standard to have well separated training, validation and test sets (with a ratios of approx. 60%, 20% and 20%, resp.). I do understand that your datasets are small. In this case, a well-known evaluation method in statistical analysis is k-fold cross-validation (which I suggest you use).

I am concerned that if this paper gets published, it will serve as a reference for others to repeat the authors evaluation protocol. Deviating from standard protocols will inevitably cast doubts on the results and conclusions. To my knowledge, I have never seen any works using a similar protocol than yours. Are there any good references I might not be aware of that use a similar protocol to define train, validation and test sets in statistical analysis?

2/ A significant part of the paper is about defining and comparing different types of "weak labels": cubic, ball and Gaussian. I shared my doubts about the relevance of this comparison, given that after thresholding, their Gaussian "weak labels" are identical to the ball "weak labels" (see my last review, comment 2.1). The authors insist that it is not:

"See equations (1), (2), (3) and (4) on page 7:

The hyperparameter t_g determines the shape of Gaussian masks. The Gaussian mask becomes a ball mask when $t_g \geq \exp(0.5) \approx 0.607$, and it becomes a cubic mask when $t_g \leq \exp(1.5) \approx 0.223$. To generate a Gaussian mask that is sufficiently different from ball/cubic masks, we choose the middle point between 0.5 and 1.5 and set $t_g = \exp(-1) \approx 0.368$ in our study."

I am sorry, but equation (4) describes a thresholded (i.e. binarized), isotropic, 3D Gaussian; which is a sphere, regardless of parameter value t_g . Maybe your implementation generates what you describe, but this is not in accordance with equation (4). Also, if the generated "Gaussian" mask ("Gau-M") is between a sphere and a cube, as shown in Suppl. Fig. 2a, you shouldn't call it Gaussian. How is your Gau-M mask any different from a truncated sphere?

I don't understand the rationale behind evaluating arbitrary shapes such as cubes and truncated spheres. In this case, why not evaluate polygonal shapes?

The authors insist that the "Gau-M" mask performs better. If so, how do you explain that a truncated sphere with sharp edges yields better performance?

In Suppl. Table 3, the authors compare the localisation and classification performances when using their different masks. They conclude that the "Gaussian" mask performs better (localisation F1: 0.94 ; classification mean F1: 0.84) than the cubic mask (localisation F1: 0.92 ; classification mean F1: 0.81) and ball mask (localisation F1: 0.93 ; classification mean F1: 0.84). In my opinion, the difference in scores is not significant. For SHREC datasets, variations to up to 5% in score values are to be expected from one identical training run to another. I believe that the score differences are due to the stochasticity of the method.

Here are some secondary, but still important concerns:

3/ In my last review (see comment 3.2), I criticised the authors for their frequent use of the term "particle quality", which they do not define clearly. The authors answer is evasive and vague:

"Conceptually, particle "quality" can be assessed by the authenticity of the particles, the accuracy of the particle coordinates, and the contribution of the particles to subtomogram averaging."

So the authors define the particle "quality" as being 3 different things at the same time:

- Authenticity of the particles: this is a binary value (true or false) and can be determined by comparison with a ground truth / expert annotations.
- The accuracy of the particle coordinate: this can be defined as the distance between detected and ground truth coordinates.
- The contribution to subtomogram averaging: how do you even measure that?

The problem is that the paper becomes less precise when using the same word for describing three different concepts. My suggestion is to stop using the term "particle quality", and use precise and quantitative terms instead. Or to give a more precise definition.

Also, I proposed to add a much needed discussion on how the resolution of an obtained subtomogram average is correlated to the performance of a particle picker (I thoroughly discussed this point in my last review, comment 3.2), but the authors chose to ignore my suggestion.

4/ In my last review (see comment 3.3), I asked how the authors chose the score thresholds for template matching and DeepFinder. Choosing these thresholds greatly influences the method performances, and hence the comparison. Their answer is rather arbitrary:

"For DeepFinder, ... a cluster size in the range of [0.1V, 2V] are selected ... for EMPIAR-100045. [0.2V, 2V] ... for EMPIAR-10499. [0.1V, 2V] ... for EMPIAR-10651. [0.1V, 4V] ... for EMPIAR-11125."

But **why** did you choose those thresholds?

"For template matching, we ... roughly estimate the cross-correlation threshold".

In order to avoid doubts, I suggest to plot recall, precision and F1-score w.r.t. the method scores. And then, for sub-sequent analysis (i.e. subtomogram averaging), to choose the threshold that maximizes the F1-score w.r.t. expert annotations.

Note: is this what you display in Figures 3b, 4b and 5b? Because you don't explain w.r.t. what parameter the precision-recall curves are being plotted.

A Point-by-Point Response to Reviewers' Comments

We thank the reviewers for their thoughtful and constructive comments. To address the concerns raised in the comments, we have performed new experiments for more rigorous characterization of DeepETPicker in comparison with competing methods. We have also revised the manuscript to improve its rigor and clarity as well as to provide more accurate and comprehensive descriptions of technical details. All changes to the manuscript have been marked in red. We address specific comments of the reviewers below.

Reviewer #2

Comments: The authors have improved the manuscript from the previous submission. They have properly addressed the reviewers' concerns on the difference between Gaussian masks and spherical masks, measuring quality with subtomogram averaging, the method's generalization ability to non-spherical structures, and applicability to structures smaller than ribosomes. The authors have addressed the minor remarks as well and made appropriate changes to the manuscript.

In my opinion, the proposed method, DeepETPicker, will be a very useful tool for the cryo-ET, as well as, the structural biology community to perform semi-automated picking of macromolecules in situ. Moreover, DeepETPicker can be used not only for picking large macromolecules like ribosome, but also for smaller non-spherical macromolecules. Consequently, I vote for accepting the manuscript unaltered in its current form.

Response: We thank the reviewer for the positive and constructive comments, which truly have helped us to improve our manuscript and have encouraged us to develop deep learning-based computational image analysis tools for high-resolution cryo-electron tomography *in situ*.

Reviewer #3

Comments: I would like to thank the authors for their work and their efforts in answering my questions. With these answers, new informations came into light, which confirm concerns I described in my first review. As a result, I am sorry to announce that I consider a major revision to be necessary. I have two main concerns:

1/ The train, validation and test sets are not clearly separated.

1.1/ During training, the procedure samples 3D patches from the tomogram set. I asked if the train and validation patches originate from different tomograms, because if not, there is a risk that a patch contains both train and validation particles (see my last review, comments 3.9 and 3.10). The authors answered:

"For all the four experimental tomogram datasets, the coordinates of the manually picked particles are sorted in the order of z, y, and x from the smallest to the largest before split of the training and validation sets. This sorting strategy reduces the risk of overlap between the training and the validation sets."

Maybe it reduces the risk, but it does not eliminate it. For a rigorous statistical analysis, no overlap between train, validation and test sets should be allowed.

Response: We are deeply grateful to the reviewer for all the careful and constructive comments, which have greatly helped us to improve the rigor and technical quality of our study and manuscript.

Specifically, we acknowledge that our previous coordinate sorting approach reduces but does not eliminate the risk of overlap. To completely eliminate the risk of overlap, for all the four experimental datasets, we performed new experiments in which we manually picked validation particles from a tomogram that differs from the tomograms used for picking training particles. Further details are summarized in **Supplementary Table A8** and **Supplementary Table A9**. We would also like to note that the new procedure of model validation caused little to no change to the results of model selection. Consequently, there was essentially no change to the subsequent results of model testing.

Comments: 1.2/ But most importantly, the authors added new information about their datasets:

"For EMPIAR-10045, 150 particles were manually labelled [...]. The coordinates of these 150 particles are sorted in order of z, y, and x from the smallest to the largest. For crYOLO, DeepFinder and DeepETPicker, the first 135 sorted particles are used for training, and the remaining 15 particles are used for

validation. In the inference stage, the trained model of each method is used particles on all tomograms."

So first, the authors divide the dataset into training and validation sets. Then, for evaluating the method, they proceed to join the training and validation sets to form their test set, which should not be allowed. They proceed in a similar way for other datasets.

Answering to my previous concerns, the authors write that for some of the datasets:

"to avoid the performance metrics being overestimated, we remove training particles from the detected particles for all methods."

So I conclude that for the dataset cited above (EMPIAR-10045), after removing the training particles, the test set is constituted of only 15 particles, which is clearly not enough to have a strong estimate of performance. And I also conclude that their test set is identical to their validation set.

To justify their protocol, the authors wrote:

"We agree with the reviewer and acknowledge that the performance metrics of all methods on these two datasets are overestimated. We have compared the results with and without a split of the training and test sets in Supplementary Figure A1. The conclusions on comparing performance of different methods do not change qualitatively."

While I understand partially that the authors proceed in this way because the sizes of their datasets are limited, this is certainly not a standard protocol for evaluating a machine learning method.

Response: We agree with the reviewer that training, validation and test sets should be rigorously separated. We have revised the manuscript to ensure the complete separation of training, validation and test sets. In the following, we summarize the changes made.

- 1) For the two simulation datasets from SHREC2020 and SHREC2021 Challenges, the training, validation, and test sets were already rigorously separated by strictly following the protocols provided by the organizers in the previous version of the manuscript. Consequently, in this version of the manuscript, no changes were made.
- 2) For the experimental datasets of EMPIAR-10045 and EMPIAR-10499, the training, validation, and test set were already well separated in the previous version of the manuscript. In this version of the manuscript, except that the validation particles were picked from different tomograms, no changes were

made. Please refer to **Supplementary Table A8** and **Supplementary Table A9** for further details.

- 3) For the experimental datasets of EMPIAR-10651 and EMPIAR-11125, the training and validation set were merged with the test set in the previous version of the manuscript. In this version of the manuscript, the training, validation and test sets were rigorously separated in calculating the precision-recall curves (**Fig. 5b** and **Supplementary Fig. 10b**). Please also refer to **Supplementary Table A8** and **Supplementary Table A9**. And the validation particles were picked from different tomograms.

It is standard to have well separated training, validation and test sets (with a ratios of approx. 60%, 20% and 20%, resp.). I do understand that your datasets are small. In this case, a well-known evaluation method in statistical analysis is k-fold cross-validation (which I suggest you use).

Response: As described above, the training, validation and test sets were rigorously separated in this version of the manuscript.

However, we did not follow the 60%-20%-20% ratio because DeepETPicker was designed to require only a small number of training particles. As shown in **Supplementary Table A8**, we generally used a much larger number of particles for testing. For example, for EMPIAR-10499, we used 106 particles for training, 11 particles for validation, and 11921 particles for testing.

We thank the reviewer for the constructive suggestion of k-fold cross-validation. However, because the training, validation and test sets were well separated in this version of the manuscript, we thought that it was no longer essential.

I am concerned that if this paper gets published, it will serve as a reference for others to repeat the authors evaluation protocol. Deviating from standard protocols will inevitably cast doubts on the results and conclusions. To my knowledge, I have never seen any works using a similar protocol than yours. Are there any good references I might not be aware of that use a similar protocol to define train, validation and test sets in statistical analysis?

Response: We understand the concerns and agree with the reviewer. As described above, we have now fully revised the manuscript by strictly following the standard protocols. We hope this resolves the concerns of the reviewer.

Specifically, for the calculation of performance metrics such as precision, recall and F1-scores, the training, validation and testing sets were rigorously separated for the two simulation datasets and the four experimental datasets. In this version of the manuscript, there is no deviation from standard protocols of machine learning or deep learning.

The only exception in this version of the manuscript is when we calculated the B-factor, global resolution, local resolution, maximum value probability, and log-likelihood distribution for two of the experimental datasets, EMPIAR-100651 and EMPIAR-11125, we combined the training and validation particles with the testing particles for reconstruction. This was mainly because the numbers of particles were very limited. However, we thought this was acceptable for two reasons. First, this was performed in the same way for all the methods compared to ensure a fair comparison. Second, this is consistent with the practice of users in real-world applications when the goal is to use the maximal number of real particles for reconstruction.

Comments: 2/ A significant part of the paper is about defining and comparing different types of "weak labels": cubic, ball and Gaussian. I shared my doubts about the relevance of this comparison, given that after thresholding, their Gaussian "weak labels" are identical to the ball "weak labels" (see my last review, comment 2.1). The authors insist that it is not:

"See equations (1), (2), (3) and (4) on page 7:

The hyperparameter tg determines the shape of Gaussian masks. The Gaussian mask becomes a ball mask when $tg \geq \exp(0.5) \approx 0.607$, and it becomes a cubic mask when $tg \leq \exp(1.5) \approx 0.223$. To generate a Gaussian mask that is sufficiently different from ball/cubic masks, we choose the middle point between 0.5 and 1.5 and set $tg = \exp(-1) \approx 0.368$ in our study."

I am sorry, but equation (4) describes a thresholded (i.e. binarized), isotropic, 3D Gaussian; which is a sphere, regardless of parameter value tg . Maybe your implementation generates what you describe, but this is not in accordance with equation (4). Also, if the generated "Gaussian" mask ("Gau-M") is between a sphere and a cube, as shown in Suppl. Fig. 2a, you shouldn't call it Gaussian. How is your Gau-M mask any different from a truncated sphere?

Response: We regret the confusion caused by our previous inaccurate term of Gaussian mask (Gau-M). Equations (1-4) on page 7 define three simplified masks, where (1) sets the domain of the masks. As pointed out by the reviewer, the Gaussian mask in our study is indeed a truncated sphere. To improve its clarity, we

have replaced the term and abbreviation “Gaussian mask (Gau-M)” with “Truncated Ball mask (TBall-M)”, “Cubic mask (Cub-M)” with “Cubic mask (Cubic-M)”, “Ball mask (Bal-M)” with “Ball mask (Ball-M)”.

I don't understand the rationale behind evaluating arbitrary shapes such as cubes and truncated spheres. In this case, why not evaluate polygonal shapes? The authors insist that the "Gau-M" mask performs better. If so, how do you explain that a truncated sphere with sharp edges yields better performance?

Response: We initially used Ball masks (Ball-M) as our weak labels and tested different radius settings. However, the experimental results showed that Ball-M with a diameter of for example 7 usually could not pick all types of particles. Often one or more types of particles were missed. This motivated us to examine weak labels with different shapes, which should be easy to implement and should have good approximations in voxels for actual particle masks. The first type of weak labels we considered were cubic masks (Cubic-M). However, because the surfaces of macromolecular particles are usually smooth, we found the regions near the edges of Cubic-M were noisy. To reduce the number of noisy voxels, we tried a new type of weak labels, Truncated Ball masks (TBall-M), which do not have the same sharp edges as cubic masks. Overall, we examined these three different types of weak labels under different radius settings. Compared to Cubic-M and Ball-M masks, TBall-M masks provided more stable and better localization and classification performance, regardless of what radius was chosen (**Fig. 2d and Supplementary Table 3**). Another more important conclusion based on our experiments was that utilizing simplified masks with constant diameters as training labels achieved comparable, if not better, performance as real segmentation masks. Furthermore, simplified masks with constant diameters avoided the issue of class imbalance and simplified the selection of loss functions (**Supplementary Methods A.5**). Because TBall-M masks consistently achieved good performance in particle picking, we did not investigate other more complex shapes such as polygons. We have added a brief summary of the rationale described above to **Supplementary Methods A.3**.

In Suppl. Table 3, the authors compare the localisation and classification performances when using their different masks. They conclude that the "Gaussian" mask performs better (localisation F1: 0.94 ; classification mean F1: 0.84) than the cubic mask (localisation F1: 0.92 ; classification mean F1: 0.81) and ball mask (localisation F1: 0.93 ; classification mean F1: 0.84). In my opinion, the difference in scores is not significant. For SHREC datasets, variations to up to

5% in score values are to be expected from one identical training run to another. I believe that the score differences are due to the stochasticity of the method.

Response: We discussed part of the rationale of choosing Truncated Ball mask above. On the SHREC simulation datasets, variations indeed exist from one identical training run to another. We found in our experiments that the variations in F1-score were generally lower than 2% for DeepETPicker. When we examined the results under different radius settings, we observed a small but consistent advantage of Truncated Ball masks in localization and classification. More importantly, compared with Cubic-M and Ball-M masks, we found that TBall-M masks consistently provided more stable and better localization and classification performance regardless of the radius settings. Taking into account all these factors, we chose Truncated Ball masks for our study.

Comments: Here are some secondary, but still important concerns:

3/ In my last review (see comment 3.2), I criticised the authors for their frequent use of the term "particle quality", which they do not define clearly. The authors answer is evasive and vague:

"Conceptually, particle "quality" can be assessed by the authenticity of the particles, the accuracy of the particle coordinates, and the contribution of the particles to subtomogram averaging."

So the authors define the particle "quality" as being 3 different things at the same time:

- Authenticity of the particles: this is a binary value (true or false) and can be determined by comparison with a ground truth / expert annotations.**
- The accuracy of the particle coordinate: this can be defined as the distance between detected and ground truth coordinates.**
- The contribution to subtomogram averaging: how do you even measure that?**

The problem is that the paper becomes less precise when using the same word for describing three different concepts. My suggestion is to stop using the term "particle quality", and use precise and quantitative terms instead. Or to give a more precise definition.

Also, I proposed to add a much needed discussion on how the resolution of an obtained subtomogram average is correlated to the performance of a particle picker (I thoroughly discussed this point in my last review, comment 3.2), but the authors chose to ignore my suggestion.

Response: We thank the reviewer for the constructive comments. The contribution to subtomogram averaging can be defined by the value of log-likelihood of the particle based on the mathematics principle behind RELION (see Journal of Molecular Biology, 415(2): 406-18, 2012), which has been used to measure the quality of particles picked by different particle pickers. For the particle that is false positive or has a large deviation of its true center, a lower log-likelihood and a lower maximum value probability would be calculated to down-weight its contribution to the final subtomogram averaging. Following the suggestion of the reviewer, to make a more rigorous statement, we have revised the manuscript and used precise and quantitative terms to replace the term “particle quality”. Specifically, we used the following specific metrics: B-factor, global resolution, local resolution, log-likelihood distribution, and maximum value probability. We have also added a discussion of the relationship between the resolution of an obtained subtomogram and the performance of a particle picker.

Specifically, following the suggestions of the reviewer, we have added the following discussion to the main text on page 12, starting from line 22.

“If the authenticity of the picked particles is worse, i.e., more false positive junk particles are picked, the SNR of the set of picked particles becomes worse. Then worse subtomogram averaging with reduced local and global resolutions is expected. Furthermore, a larger number of particles would be needed to reach the same reconstruction resolution. Thus a higher B-factor would be expected in case the local and global reconstruction resolutions may not be sensitive enough. It should be noted that if there is conformational heterogeneity in the specimen, the reconstruction resolutions, either local or global, may not be a good indicator to evaluate different pickers. More rigorous investigations using e.g., map inspection and 3D classification are needed.

More importantly, the authenticity and the coordinates accuracy of the picked particles can be assessed by the distributions of particle log-likelihood and maximum value probability. For a picked particle that is false positive or has a large deviation from its true centre, a lower log-likelihood and a lower maximum value probability would be calculated to down-weight its contribution to the final subtomogram averaging. Therefore, larger number of particles with higher log-likelihood and maximum value probability indicate better coordinates accuracy of the picked particles and better authenticity of the picked particle set.”

Following the suggestion of the reviewer, we have also included the following revised discussion to the main text on page 27, starting from line 1.

“We also examined the performance of DeepETPicker on four experimental datasets (EMPIAR-10045, EMPIAR-10651, EMPIAR-10499, and EMPIAR-11125). We developed multiple particle metrics to compare the performance of DeepETPicker with that of other methods. We found that the particles picked by DeepETPicker consistently showed the best authenticity and coordinates accuracy with the highest log-likelihood contributions and the highest cumulative ratio of particles versus the maximum value probability, which was consistent with the observation that the particles picked by DeepETPicker produced reconstruction maps with the best global resolution, the best local resolution and the smallest B-factors. Although the assessment of reconstruction resolutions may be affected by the potential existence of specimen conformational heterogeneity, we found when comparing DeepETPicker with other methods such as crYOLO, DeepFinder and TM, the particles not picked by DeepETPicker but selected by other methods generally failed to produce correct reconstructions. Therefore, the extensive analyses suggested that the accuracy and precision of the particles picked by DeepETPicker were substantially better than those of the other methods.”

Comments: 4/ In my last review (see comment 3.3), I asked how the authors chose the score thresholds for template matching and DeepFinder. Choosing these thresholds greatly influences the method performances, and hence the comparison. Their answer is rather arbitrary:

"For DeepFinder, ... a cluster size in the range of [0.1V, 2V] are selected ... for EMPIAR-100045. [0.2V, 2V] ... for EMPIAR-10499. [0.1V, 2V] ... for EMPIAR-10651. [0.1V, 4V] ... for EMPIAR-11125."

But *why* did you choose those thresholds?

"For template matching, we ... roughly estimate the cross-correlation threshold".

In order to avoid doubts, I suggest to plot recall, precision and F1-score w.r.t. the method scores. And then, for sub-sequent analysis (i.e. subtomogram averaging), to choose the threshold that maximizes the F1-score w.r.t. expert annotations.

Note: is this what you display in Figures 3b, 4b and 5b? Because you don't explain w.r.t. what parameter the precision-recall curves are being plotted.

Response: We regret that our previous writing caused confusion. Overall, the purpose of setting these thresholds is to filter out false-positive particles for fair performance comparison. They were chosen interactively by following well-defined procedures described below.

We thank the reviewer for the constructive suggestion of choosing the threshold that maximizes the F1-score. However, we respectfully disagree here. Because expert annotations are generally not available for real-world applications, it would not be practical to set the threshold in this way. Although the way we used to determine the thresholds was empirical, it followed well-defined and principled procedures. In the following, we provide a detailed description of the rationale and procedures for setting these thresholds. We have now added this description as **Supplementary Methods A.7**.

- For DeepFinder, we performed model training, segmentation and clustering by following the tutorial (<https://deepfinder.readthedocs.io/en/latest/>) provided by its developers. When we examined the initial results provided by DeepFinder, we found that the number of particles it detected was much higher than the number of manually labeled particles. When we plotted the picked particles back to the tomogram, we found that there were many false positives. If we used the particles as detected by DeepFinder for sub-sequent analysis, its performance would be unfairly underestimated. For DeepFinder, the voxels of the cryo-ET tomogram were first classified into N classes. Then the multi-class voxel-wise classification map was spatially clustered into 3D connected components, with each cluster corresponding to a unique particle. In the original paper of DeepFinder, it was written that “Clusters that are significantly smaller than the size of target particles are considered as false positives and are discarded”. Thus, for a fair comparison with DeepFinder, we interactively adjusted the volume threshold as 0-20% the size of target particles based on visual inspection. Indeed, when we compared the F1-scores of DeepFinder with and without setting the volume thresholds on the testing set, we found that for the three experimental datasets the F1-score consistently improved after setting the thresholds (**Supplementary Table A6** and **Supplementary Figure A1**).
- For template matching, we used mainly the template matching function “dynamo_match”. There are two parameters that may affect particle selection. The parameter 'cr' (cone range) defines orientations that will be searched for

inside a cone. In our experiment, we used the most typical value of 360 (sample the full sphere). The parameter 'cs '(cone sampling) determines the scanning density inside the sphere. In our experiment, we used the most typical value of 30 (sample the full sphere). The function generates tbl-format table files, where the tenth column shows the cross-correlation coefficient. For each tomogram, we obtained a plot of the cross-correlation values found on the local maxima of the cc volume with the order. Given that the cross-correlation values of the peaks appeared in an ascending order, we interactively checked the quality of the peaks by clicking on the curve to select one particle and then selecting related visualization option. We clicked on a few particles within the kink region to interactively estimate the cross-correlation threshold (**Supplementary Table A7**). Finally, the coordinates TM picked were plotted back to tomogram for further verification.

Each of the four methods (DeepETPicker, crYOLO, DeepFinder, and TM) have a confidence metric for its picked particles. For DeepETPicker and DeepFinder, the voxels of the cryo-ET tomogram are classified into N classes. The confidence of a particle is measured by the volume of voxel-wise classification map belonging to this particle. For crYOLO, each detected particle has a confidence metric provided in the result file directly. This confidence metric denotes the probability that the detected particle is an authentic particle. For TM, the confidence of a particle is measured by the cross-correlation coefficient. For a fair comparison between different methods, we sorted the particles of each method based on its confidence metric from the highest to the lowest. Using the manual annotation as the reference, the precision and recall of particles with confidence larger than different threshold were calculated. The precision-recall curves of different methods were then plotted together for performance comparison (**Figures 3b, 4b, 5b and Supplementary Fig. 10b**). This would eliminate the influence of manual setting of confidence threshold on the performance comparison of different methods. In the **Supplementary Methods A.9**, we have provided a detailed explanation of the parameters with respect to the precision-recall curves in **Figures 3b, 4b, 5b and Supplementary Fig. 10b**.

Reviewer #4 (Remarks to the Author):

Summary:

DeepETPicker is a CNN-based particle picker that is claimed to outperform existing Deep Learning approaches in terms of F1-score, speed, label-efficiency, and accuracy. Key building blocks of the method are a 3D ResU-Net architecture with adjustments like pyramid inputs and coordinated convolution, and a mean-pooling non-maximum-suppression module for post-processing. The authors validate their method using various synthetic and experimental datasets. Hereby, they compute F1-scores for the comparison with ground truth positions, as well as metrics captured from protein reconstructions using subtomogram averaging.

Originality and significance: The idea of training a 3D U-Net to segment protein locations (or spheres around them) has already been implemented in previous work (e.g. DeepFinder, VP-Detector). However, DeepETPicker seems to have made some effective adjustments to the architecture and post-processing, leading to more accurate predictions and a faster and much smaller model.

Since DeepETPicker is packaged with a GUI, its access is facilitated to the Cryo-ET community and may be used in experimental projects.

Data & methodology: The approach has been validated exhaustively on multiple datasets showing different proteins.

Appropriate use of statistics and treatment of uncertainties: Provided comparison values (e.g. F1-scores) seem to be based only on a single training run. The differences in some design choices (e.g. "cubic" masks vs "truncated sphere" masks only differ by few %) may not be significant. Results should, therefore, better be reported as means and standard deviations as common for computer vision papers (see Main suggested improvements for detailed suggestions).

Conclusions: The authors conclude that their approach is more accurate, faster and more label-efficient than other approaches like DeepFinder, which is corroborated by their consistently better statistics in all experiments.

Main critique and suggested improvements:

Experiments should be performed multiple times to receive significant results.

For SHERC challenge, one can indeed use cross-validation as suggested by Reviewer #3 last round of comments.

For experimental data where authors use a relatively small training data and a large testing data, there might be a batch-effect related to the training data (it becomes more severe when the training data is small). So one can fix a testing set and do e.g. five randomly-sampled training set and therefore obtain five testing results to obtain a realistic estimation of the robustness of the model performance.

Minor critique:

It's not clear to me how classification F1-score and detection F1-score are calculated in this setting: Is the detection F1-score simply throwing all protein positions together?

Is the classification F1-score the mean over all F1 scores for all classes?

Figure 2: Why are some experiments performed with Shrec2021 data and some with Shrec 2020?

I don't see a reason why one experiment makes more sense for one of them

Figure 3c: Is comparing the shapes of the log-likelihood contributions really an objective measure?

It depends very much on the resulting subtomogram average. (probably constantly high values indicate that detected subvolumes align very well)

Equation 3) and 4): Why describe a sphere once with the L2-norm and once with an exponential kernel? In my opinion, that makes equation 4 look more complicated than it is, and requires the non-intuitive calculation of $t_g=0.368$. Equation 4) can also be simply described like equation 3), but with another radius to achieve the truncation.

What loss function was used to train the multi-class experiments (i.e. Shrec challenges)? Only Dice-Loss was introduced, which, by default, is a binary classification loss.

Was a multi-class extension of Dice, like Tversky loss, used?

Overall, the authors incorporated the criticized points in their manuscript very well:
Firstly, they performed the training/validation/test splits more cleanly and clearly. The only debatable thing hereby is the merging of training, validation and test particles for subtomogram averaging and computing statistics from these averages. While the authors mention that this is also the case for compared methods, this can still distort the statistics, e.g. depending on the number of detected particles in the test set.
Second, the discussion about equation 4) seems to be displayed correct in the manuscript now, but, as mentioned above, equation 4) is still confusing and should be facilitated.
Lastly, Reviewer #3 mentioned that the "better" results of the truncated sphere vs. other weak labels may not be significant. I agree with this opinion and think that the significance could be shown by multiple model training, and showing that the means do differ with a low standard deviation, as is common practice in deep learning benchmarks.

A Point-by-Point Response to Comments of Reviewer #4

We thank the reviewer for the thoughtful and constructive comments. To address the concerns raised in the comments, we have performed new experiments with multiple training runs for more rigorous statistical characterization of DeepETPicker. We have also revised the manuscript to improve its rigor and clarity as well as to provide more accurate and comprehensive descriptions of technical details. All changes to the manuscript have been marked in red. We address specific comments of the reviewer below.

Reviewer #4

Comments:

Summary: DeepETPicker is a CNN-based particle picker that is claimed to outperform existing Deep Learning approaches in terms of F1-score, speed, label-efficiency, and accuracy. Key building blocks of the method are a 3D ResU-Net architecture with adjustments like pyramid inputs and coordinated convolution, and a mean-pooling non-maximum-suppression module for post-processing. The authors validate their method using various synthetic and experimental datasets. Hereby, they compute F1-scores for the comparison with ground truth positions, as well as metrics captured from protein reconstructions using subtomogram averaging.

Originality and significance: The idea of training a 3D U-Net to segment protein locations (or spheres around them) has already been implemented in previous work (e.g. DeepFinder, VP-Detector). However, DeepETPicker seems to have made some effective adjustments to the architecture and post-processing, leading to more accurate predictions and a faster and much smaller model.

Since DeepETPicker is packaged with a GUI, its access is facilitated to the Cryo-ET community and may be used in experimental projects.

Data & methodology: The approach has been validated exhaustively on multiple datasets showing different proteins.

Appropriate use of statistics and treatment of uncertainties: Provided comparison values (e.g. F1-scores) seem to be based only on a single training run. The differences in some design choices (e.g. “cubic” masks vs “truncated sphere” masks only differ by few %) may not be significant. Results should, therefore, better be reported as means and standard deviations as common for computer vision papers (see Main suggested improvements for detailed suggestions).

Conclusions: The authors conclude that their approach is more accurate, faster and more label-efficient than other approaches like DeepFinder, which is corroborated by their consistently better statistics in all experiments.

Response: We thank the reviewer for the thoughtful summary and constructive comments.

Specifically, regarding the concerns raised on “Appropriate use of statistics and treatment of uncertainties”, we have conducted new experiments with multiple training runs for more rigorous statistical characterization of DeepETPicker. The results are reported below in responses to specific questions.

Comments:

Main critique and suggested improvements:

Experiments should be performed multiple times to receive significant results.

For SHERC challenge, one can indeed use cross-validation as suggested by Reviewer #3 last round of comments.

Response: We thank the reviewer for the constructive comments and suggestions. Following the suggestions, we have conducted two new groups of experiments with multiple training runs to check whether the performance differences observed in our study between different particle picking models are statistically significant.

- 1) In the first group of experiments, we followed suggestions of both Reviewer #3 and Reviewer #4 and performed 10-fold cross-validation experiments to characterize performance of DeepETPicker on the SHREC2021 dataset, which consists of 10 tomograms. For each experiment, we randomly selected 8 tomograms for training, 1 tomogram for validation, and 1 tomogram for testing. Overall, performance of DeepETPicker varies in the experiments (**Supplementary Figure A2**), presumably because of the different settings of simulation parameters such as defocus levels, electron doses, and particle type compositions in generating these tomograms, as reported by the organizers of the challenge (Gubins *et al*, SHREC 2021: Classification in cryo-electron tomograms, Eurographics Proceedings, 2021). For example, we observed that DeepETPicker generally achieves lower classification and localization F1-scores on tomograms with lower signal-to-noise ratios.
- 2) Although the experiments in the first group provide insights into the performance of DeepETPicker, the results cannot be compared directly with

results of those methods from the SHREC2021 Challenge. This is because the training, validation, and test sets were partitioned following the protocol provided by the organizers. To generate results that can be used to compare DeepETPicker with the methods from SHREC2021, we performed a second group of experiments in which we followed the same protocol of partitioning training/validation/test sets and conducted the experiments 10 times using 10 different random seeds. Overall, we found that the variations of both localization F1-scores and classification F1-scores of DeepETPicker in the experiments are small, with a standard deviation of approximately 0.003~0.005 (**Supplementary Figure A2** and **Supplementary Table A10**). Because only a single F1-score is provided by SHREC2021 for each method without its statistical distribution, it is not feasible to perform direct statistical performance comparison of DeepETPicker versus these methods. However, given the observed low level of variations in F1-scores, the observed performance difference in e.g. **Supplementary Table 4**, **Supplementary Table 5**, **Supplementary Figure A3** and **Figure 2e** are generally much higher than 0.003~0.005 and therefore are likely significant. We note that it is common in deep learning studies to compare performance of competing methods without using explicit statistical tests.

For experimental data where authors use a relatively small training data and a large testing data, there might be a batch-effect related to the training data (it becomes more severe when the training data is small). So one can fix a testing set and do e.g. five randomly-sampled training set and therefore obtain five testing results to obtain a realistic estimation of the robustness of the model performance.

Response: For EMPIAR-10045 and EMPIAR-10499, we used a relatively small training set and a large testing set. We agree with the reviewer that when a relatively small training dataset and a large testing dataset are used, there can be a risk of batch-effect related to the training data.

To obtain a realistic estimation of the robustness of the model performance, we followed suggestions of the reviewer and fixed the validation set and the test set. We then randomly sampled five training sets to obtain five testing results on EMPIAR-10045 and EMPIAR-10499. Details on partitioning the training/validation/testing sets are summarized in **Supplementary Table A11**. Specifically, EMPIAR-10045 consists of 7 tomograms. Five training sets and one validation set are randomly sampled from tomograms labeled *tomo0* to *tomo3*. And tomograms labeled *tomo4* to *tomo6* are used for testing. EMPIAR-

10499 consists of 10 tomograms. Five training sets and one validation set are randomly sampled from tomograms labeled *tomo0* to *tomo5*. And tomograms labeled *tomo6* to *tomo9* are used for testing. In this way, five testing results are obtained for both EMPIAR-10045 and EMPIAR-10499.

- For the EMPIAR-10045 dataset, three methods, i.e., DeepETPicker, DeepFinder and crYOLO, were trained by the five training sets and tested on the same test set. In terms of picking performance, compared with DeepFinder and crYOLO, DeepETPicker provides more consistent localization F1-score with the highest mean and the lowest standard deviation (**Supplementary Figure A4.a**). In terms of inference time, the mean for DeepETPicker is 62 seconds, which is 25 times faster than DeepFinder and 2.5 times faster than crYOLO (**Supplementary Figure A4.b**).
- For the EMPIAR-10499 dataset, two methods, i.e., DeepETPicker and crYOLO were trained by five training sets and tested on the same test set. Training of DeepFinder failed to converge in multiple cases, as it generally requires substantially more particles for training. Consequently, performance metrics could not be reported for DeepFinder. In terms of picking performance, DeepETPicker provides much more consistent localization F1-score with higher mean and lower standard deviation than crYOLO (**Supplementary Figure A4.c**). In terms of inference time, the mean for DeepETPicker is 108 seconds, which is comparable to 103 seconds for crYOLO (**Supplementary Figure A4.d**).

Overall, these new experiments and statistical analyses confirmed the robustness of DeepETPicker performance. We have added detailed descriptions of these results to **Supplementary Methods A.11**.

Comments: Minor critique:

It's not clear to me how classification F1-score and detection F1-score are calculated in this setting: Is the detection F1-score simply throwing all protein positions together?

Is the classification F1-score the mean over all F1 scores for all classes?

Response: Localization F1-score and classification F1-score are two main performance metrics used in the SHREC challenges. Detection F1-score is the same as "Localization F1-score" in the SHREC challenges. Each protein in a given

tomogram occupies certain voxels. If the coordinate of a predicted particle is within the voxels of a protein, this protein is considered to be detected according to the rules and code provided by the organizers of the challenge. See the official code of SHREC challenges (<https://www.shrec.net/cryo-et/>) for further details. The number of all detected proteins in a tomogram is counted as the number of true positives (TPs). Then the detection F1-score can be calculated.

For each class of proteins, a classification F1-score can be obtained. Specifically, each protein in the tomogram belongs to a unique category. If the coordinate of a predicted particle is within the voxels of a protein, then the class of predicted particle is used as the predicted label of the protein. If multiple predicted particles are within the voxels of a protein, the class of the first predicted particle is used as the predicted label of the protein according to the rules and code provided by the organizers of the challenge. Based on the ground truth of all proteins and their predicted labels, a confusion matrix is obtained. Then the classification F1-score for each class of protein can be calculated. The mean over all F1-scores for all classes is used as the mean classification F1-score in this study.

Comments: Figure 2: Why are some experiments performed with Shrec2021 data and some with Shrec 2020? I don't see a reason why one experiment makes more sense for one of them

Response: SHREC 2021 and SHREC 2020 are the two most commonly used simulated datasets, aiming to benchmark different particle picking methods for localization and classification of biological macromolecules in cryo-ET tomograms. Compared to SHREC 2020, SHREC 2021 substantially improved the data generation process by incorporating e.g. multi-slice methods, varied defocusing and electron doses, as well as Fourier scaling to experimental images and by introducing membranes as an additional semantic class. Therefore, we generally choose to perform our experiments on the SHREC2021 dataset.

The only exception is in **Figure 2g**, where the influence of different numbers of training tomograms on the classification performance is examined with SHREC 2020 data. This is because DeepFinder only provides corresponding results on the SHREC2020 dataset. For a fair comparison with DeepFinder, we chose to use the results provided by DeepFinder directly. Therefore, a comparison between DeepETPicker and DeepFinder was performed using the SHREC 2020 dataset.

Comments: Figure 3c: Is comparing the shapes of the log-likelihood contributions really an objective measure? It depends very much on the resulting subtomogram

average. (probably constantly high values indicate that detected subvolumes align very well)

Response: We agree with the reviewer that the log-likelihood contributions depend on the accuracy of sub-volumes alignment, and that constantly high values indicate that detected sub-volumes align well. However, the accuracy of sub-volume alignment is actually highly correlated with the authenticity and accuracy of particle picking. Therefore, we propose to use this evaluation criterion because it can measure the accuracy of our particle picking by characterizing whether the coordinates of the picked particle centers are sufficiently accurate.

Comments: Equation 3) and 4): Why describe a sphere once with the L2-norm and once with an exponential kernel? In my opinion, that makes equation 4 look more complicated than it is, and requires the non-intuitive calculation of $t_g=0.368$. Equation 4) can also be simply described like equation 3), but with another radius to achieve the truncation.

Response: We thank the reviewer for the constructive comment. We have now rewritten equation 4 in a form similar to equation 3 with a different radius to indicate the truncation. We agree that this makes the formulation clearer and more intuitive to follow. Please see line 27 on page 7.

Comments: What loss function was used to train the multi-class experiments (i.e. Shrec challenges)? Only Dice-Loss was introduced, which, by default, is a binary classification loss.

Was a multi-class extension of Dice, like Tversky loss, used?

Response: The Dice-Loss used in our manuscript is indeed a multi-class extension of the two-class Dice, similar to the Tversky loss. We used the formulation first reported in Reference 24 (cited on line 7, page 9) but added a small value ϵ to the numerator and the denominator for numerical stability. We have revised the text to highlight the multi-class aspect of this loss explicitly. Please see line 6 on page 9.

Comments: Overall, the authors incorporated the criticized points in their manuscript very well:

Firstly, they performed the training/validation/test splits more cleanly and clearly. The only debatable thing hereby is the merging of training, validation and test particles for subtomogram averaging and computing statistics from these averages. While the authors mention that this is also the case for compared methods, this can still distort the statistics, e.g. depending on the number of detected particles in the test set.

Response: We agree with the reviewer and acknowledge that the merging of training, validation, and test particles for subtomogram averaging and computing

statistics such as B-factor, global resolution, local resolution, and log-likelihood distribution from these averages is not the optimal solution. When sufficient numbers of particles are available, we partitioned the training/validation/test sets strictly. For the two experimental datasets EMPIAR-10045 and EMPIAR10499, we only used the test set for subtomogram averaging and statistics computing.

However, when the numbers of particles are very limited, using only the test set for subtomogram averaging and statistics computing may generate unstable and inaccurate results. For EMPIAR-10651 and EMPIAR-11125, because the total numbers of particles are small, we used all the particles for subtomogram averaging and statistics computing. We note that in practice, after picking particles, researchers often use the entire set of particles to maximize the number of particles used for subtomogram averaging and statistics computing. Still, we acknowledge the limitation of our study in **Supplementary Methods A.8** line 6 from the bottom of the last paragraph.

Comments: Second, the discussion about equation 4) seems to be displayed correct in the manuscript now, but, as mentioned above, equation 4) is still confusing and should be facilitated.

Response: We thank the reviewer for the constructive comment. We have now rewritten equation 4 similarly as equation 3 to make the formulation clearer and more intuitive to follow. Please see line 27 on page 7.

Comments: Lastly, Reviewer #3 mentioned that the “better” results of the truncated sphere vs. other weak labels may not be significant. I agree with this opinion and think that the significance could be shown by multiple model training, and showing that the means do differ with a low standard deviation, as is common practice in deep learning benchmarks.

Response:

a) Comparison of performance of different types of weak labels via cross-validation experiments

Following suggestions by both Reviewer #3 and Reviewer #4, we performed 10-fold cross-validation experiments on the SHREC2021 dataset, which consists of 10 tomograms, to compare the performance of the three types of simplified (weak) labels: TBall-M, Ball-M and Cubic-M. We randomly selected 8 tomograms for training, 1 tomogram for validation, and 1 tomogram for testing. Overall, we found no statistically significant differences between the three types of weak labels in terms of their mean classification and localization F1-scores (**Supplementary**

Figure A3.a). However, compared to Ball-M and Cubic-M labels, TBall-M masks provide more stable performance (**Supplementary data.xlsx**). Specifically, in all experiments, TBall-M picked all types of particles. In 4 out of 10 experiments, the Ball-M mask failed to pick all types of particles, missing either one or more types of particles. In 1 out of 10 experiments, the Cubic-M mask failed to pick all types of particles.

b) Comparison of performance of different types of weak labels on the same training/validation/test datasets with different randomization seeds

The results of 10-fold cross-validation cannot be compared with the reported results of the SHREC2021 Challenge because the training, validation, and test sets were partitioned following the protocol provided by the organizers. To generate results that can be compared with the methods of SHREC2021, we followed the same protocol of partitioning training/validation/test sets and performed multiple times of experiments by using 10 different random seeds. Compared to Ball-M and Cubic-M masks, TBall-M mask provides more consistent localization and classification performances with the highest mean and the lowest standard deviation (**Supplementary Figure A3** and **Supplementary Table A10**). Specifically, for the classification F1-score, the standard deviation of TBall-M is 0.005, while the standard deviation for Ball-M and Cubic-M masks are 0.021 and 0.032, respectively. For the localization F1-score, the standard deviation of TBall-M is 0.003, while the standard deviation for Ball-M and Cubic-M masks are 0.014 and 0.013, respectively.

Furthermore, using randomized two-sample t-test (rndtttest2) and ranksum test, we found significant difference between TBall-M and Ball-M in term of classification F1-score, with a p-value of 0.027 for rndtttest2 and 0.023 for ranksum (**Supplementary Figure A3.b**). There is also significant difference between TBall-M and Ball-M in term of localization F1-score, with a p-value of 0.043 (**Supplementary Figure A3.c**). Besides, similar to the conclusion on 10-fold cross-validation experiments, TBall-M masks provide more stable classification performance than Ball-M and Cubic-M masks (**Supplementary data.xlsx**). Specifically, TBall-M picked all types of particles in all experiments. In 1 out of 10 experiments, Ball-M mask failed pick all types of particles. In 2 out of 10 experiments, Cubic-M mask failed pick all types of particles.

Description of the cross-validation experiments has been added to **Supplementary Methods A.10**. See also **SupplementaryData.xlsx** for further details.

In the previous version of the manuscript, we wrote “Compared to Cubic-M and Ball-M masks, TBall-M masks provide more stable and better localization and classification performance regardless of which diameter setting method is chosen”. We have now revised the text as “**Compared to Cubic-M and Ball-M masks, TBall-M masks provide more stable and consistent localization and classification performance**” on page 18 line 1.

Reviewer #4 (Remarks to the Author):

The authors followed our recommendations to include a variance analysis of their results using multiple training runs. Particularly, they have added standard deviations to their results on both ablation study and comparison with competing methods.

Their results now show a clearer picture and more clearly highlight the training robustness of the truncated sphere as design choice, as well as the difference to other methods.

While the batch effect of sampling training data does seem to play a role in the model's performance on the test set, DeepETPicker seems to still consistently perform better than DeepFinder and Cryolo.

The authors also incorporated our minor points of potential improvements.

Reviewer #4 (Remarks on code availability):

I only check the code availability (it is indeed in the github) and instructions seems to be clear to run the code. But I did not run the code myself.

A Point-by-Point Response to Comments of Reviewer #4

Reviewer #4

Comments: The authors followed our recommendations to include a variance analysis of their results using multiple training runs. Particularly, they have added standard deviations to their results on both ablation study and comparison with competing methods.

Their results now show a clearer picture and more clearly highlight the training robustness of the truncated sphere as design choice, as well as the difference to other methods.

While the batch effect of sampling training data does seem to play a role in the model's performance on the test set, DeepETPicker seems to still consistently perform better than DeepFinder and Cryolo.

The authors also incorporated our minor points of potential improvements.

Response: We thank the reviewer for the positive and constructive comments. The results presented in the paper come from years of careful work in the hope of providing a fast and accurate tool for automated picking of 3D particles for high-resolution cryo-electron tomography *in situ*.

Comments: (Remarks on code availability):

I only check the code availability (it is indeed in the github) and instructions seems to be clear to run the code. But I did not run the code myself.

Response: We thank the reviewer for the careful comments. We provide two methods for deployment of DeepETPicker: Docker and Conda. We have tested the code carefully and fully expect the code to run normally by following the instructions provided.